CERN-TH-2020-162

# Applications of dispersive sum rules:
# $\epsilon$-expansion and holography

Dean Carmi[1,2], Joao Penedones[1], Joao A. Silva[1], Alexander Zhiboedov[3]

[1] *Fields and Strings Laboratory, Institute of Physics, École Polytechnique Fédérale de Lausanne (EPFL)*
*CH-1015 Lausanne, Switzerland*
[2] *Department of Mathematics and Physics University of Haifa at Oranim, Kiryat Tivon 36006, Israel*
[3] *CERN, Theoretical Physics Department, 1211 Geneva 23, Switzerland*

**Abstract**

We use Mellin space dispersion relations together with Polyakov conditions to derive a family of sum rules for Conformal Field Theories (CFTs). The defining property of these sum rules is suppression of the contribution of the double twist operators. Firstly, we apply these sum rules to the Wilson-Fisher model in $d = 4 - \epsilon$ dimensions. We re-derive many of the known results to order $\epsilon^4$ and we make new predictions. No assumption of analyticity down to spin 0 was made. Secondly, we study holographic CFTs. We use dispersive sum rules to obtain tree-level and one-loop anomalous dimensions. Finally, we briefly discuss the contribution of heavy operators to the sum rules in UV complete holographic theories.

# 1 Introduction and Summary

Mellin space is a natural arena to perform computations in conformal field theories [1–4]. In [5] Mellin amplitudes were studied in a nonperturbative setting, and their main properties, such as existence, analyticity and polynomial boundedness, were established. It was also found that Mellin space is extremely convenient to write down nonperturbative Conformal Field Theory (CFT) dispersion relations which when combined with Polyakov conditions, or equivalently consistency with the Operator Product Expansion (OPE), lead to various sum rules with interesting properties. There are other ways to derive similar sum rules in coordinate space directly [6,7]. Recently, in [8] equivalence between various approaches to dispersion relations in CFTs was established, providing

bridges between various methods and allowing to take advantage of each based on the problem at hand. In the present paper we use functionals derived using dispersion relations in Mellin space to study the four-point function of the fundamental scalar field $\phi$ in the Wilson-Fisher (WF) model in $d = 4 - \epsilon$ dimensions [9]. We reproduce and confirm previously known results up to order $\epsilon^4$, as well as derive new results. We also consider perturbative scalar field theories in AdS and provide further nontrivial tests of the dispersive Mellin sum rules and their utility.

The $\epsilon$-expansion of the Wilson-Fisher model has been studied using conformal bootstrap techniques previously [10–13]. Most notably, in [14–16] a bootstrap scheme based on the sum of Polyakov blocks in the three channels was proposed. This method was used to obtain CFT data up to order $\epsilon^3$. It is unclear whether such a method holds nonperturbatively, or if it can be used in the Wilson-Fisher model to extract predictions to higher order in $\epsilon$ [8]. In [17] the $\epsilon$-expansion was studied to order $\epsilon^4$ using the Lorentzian inversion formula and large spin re-summation.

In section 3, we use dispersive Mellin sum rules to derive the OPE data of low twist operators perturbatively in the $\epsilon$-expansion. We confirm the predictions of [14,17]. Our procedure is systematic and no assumptions of analyticity down to spin 0 were made. Furthermore, we make some new predictions. These are

- The averaged OPE coefficients of twist 4 operators at order $\epsilon^3$, see table 2.

- The coefficient of $\phi^2$ in the $\phi \times \phi$ OPE at order $\epsilon^4$, see table 3.

It would be very interesting to develop the methods of this paper further. This can potentially enable the computation of CFT data at order $\epsilon^5$ and higher. Doing this requires a better handle on the various sums and integrals that involve Mack polynomials that we discuss below. We leave this for future work.

In section 4, we consider applications of dispersive sum rules to perturbative field theories in Anti-de Sitter (AdS) space. We study scalar fields with quartic and cubic interactions at tree and 1-loop level. In this context, our main technical result is formula (4.12) for the 1-loop anomalous dimension of the leading double-twist operators in $\lambda\phi^4$ theory. We also discuss the implications of dispersive Mellin sum rules for UV complete holographic CFTs, *i.e.* dual to quantum gravity in AdS. We explain that the contribution of heavy operators is enhanced relative to the naive perturbative expansion. This general mechanism is discussed in more detail for the theory of a derivatively coupled scalar (see section 4.3). It would be interesting to understand precisely how heavy operators contribute to the dispersive sum rules in CFTs dual to weakly coupled AdS gravitational theories.

The plan of the paper is the following. In section 2, we briefly review the derivation of sum rules from dispersion relations in Mellin space. We introduce a family of sum rules and test them with Mean Field Theory (MFT). Sections 3 and 4 are the main part of the paper and were described above. We include appendices: A on Mack polynomials, B on the known CFT data of the WF fixed point and C with some auxiliary formulas.

## 2 Sum rules from dispersion relations in Mellin space

We consider the four-point function of a scalar primary operator $\mathcal{O}$ of scaling dimension $\Delta$. The Mellin amplitude $M(\gamma_{12}, \gamma_{14})$ encodes the connected part of the correlator via the double integral

$$
\langle \mathcal{O}(x_1)\mathcal{O}(x_2)\mathcal{O}(x_3)\mathcal{O}(x_4)\rangle_{\text{conn}} = \frac{1}{\left(x_{13}^2 x_{24}^2\right)^\Delta} \int_\mathcal{C} \frac{d\gamma_{12} d\gamma_{14}}{(2\pi i)^2} \left(\frac{x_{12}^2 x_{34}^2}{x_{13}^2 x_{24}^2}\right)^{-\gamma_{12}} \left(\frac{x_{14}^2 x_{23}^2}{x_{13}^2 x_{24}^2}\right)^{-\gamma_{14}}
$$

$$
\times \ \Gamma(\gamma_{12})^2 \Gamma(\gamma_{14})^2 \Gamma(\Delta - \gamma_{12} - \gamma_{14})^2 M(\gamma_{12}, \gamma_{14}),
$$

where the integration contour $\mathcal{C}$ guarantees that the OPE in every channel is correctly reproduced. Crossing symmetry of the correlator implies that

$$
M(\gamma_{12}, \gamma_{14}) = M(\gamma_{14}, \gamma_{12}) = M(\gamma_{12}, \gamma_{13}), \quad \gamma_{12} + \gamma_{13} + \gamma_{14} = \Delta. \tag{2.1}
$$

Mellin space dispersion relations are analogous to the scattering amplitude dispersion relations and allow (upon subtractions) one to express the Mellin amplitude in terms of its discontinuity, which is given by the OPE data of the correlator [5]. Equivalently, in coordinate space one can reconstruct the correlation function from its double discontinuity [6,7].

Let us briefly review the derivation of sum rules from Mellin space. Firstly, one assumes the Regge bound [5]

$$
\lim_{|\gamma_{12}| \to \infty} \frac{M(\gamma_{12}, \gamma_{13})}{|\gamma_{12}|^{1+\delta}} = 0, \qquad \text{Re}\,\gamma_{13} > \Delta - \frac{\tau_{gap}}{2}, \qquad \delta > 0, \tag{2.2}
$$

where $\tau_{gap}$ is the minimal twist operator different from the unit operator that appears in the OPE of $\mathcal{O} \times \mathcal{O}$. The Regge bound (2.2) is supposed to hold in any unitary CFT.[1] In some theories, called transparent in [18], the Regge bound above holds also for $\delta = 0$ because the nonperturbative Regge intercept obeys $j_0 < 1$. This seems to be the case for the 3d Ising CFT [18], and the 3d $O(2)$ CFT [19]. In principle, one can take advantage of this property to derive more sum rules but we do not do it in this paper.

Then, our sum rules follow from

$$
\omega_F \equiv \oint_{\mathcal{C}_\infty} \frac{d\gamma_{12}}{2\pi i} M(\gamma_{12}, \gamma_{13}) F(\gamma_{12}, \gamma_{13}) = 0, \tag{2.3}
$$

where the contour $\mathcal{C}_\infty$ encircles $\gamma_{12} = \infty$ and the rational function $F$ decays at least as fast as $1/\gamma_{12}^3$ at large $\gamma_{12}$. Different choices of the function $F$ and different values of $\gamma_{13}$ lead to different sum rules after closing the integration contour using Cauchy's theorem and

$$
\text{Res}_{\gamma_{12} = \Delta - \frac{\tau}{2} - m} M(\gamma_{12}, \gamma_{13}) = -\frac{1}{2} C_{\tau,\ell}^2 \mathcal{Q}_{\ell,m}^{\tau,d}(-2\gamma_{13}), \tag{2.4}
$$

where $m$ is a non-negative integer, $C_{\tau,\ell}^2$ is the square of the OPE coefficient of the operator $\mathcal{O}_{\tau,\ell}$ with twist $\tau \equiv \Delta_{\mathcal{O}_{\tau,\ell}} - \ell$ and spin $\ell$ that appears in the OPE of $\mathcal{O} \times \mathcal{O}$, and the Mack polynomials $\mathcal{Q}_{\ell,m}^{\tau,d}(-2\gamma_{13})$ can be found in appendix A.

It is convenient to choose $F$ to be a rational function with poles at special locations where the Mellin amplitude vanishes (this will be made precise in the next section below).[2] Such choices lead to sum rules that only involve the CFT data ($\tau$ and $C_{\tau,\ell}$) and not the Mellin amplitude itself.

---

[1] Similarly, it is supposed to hold for correlators that come from unitary QFTs in AdS.

[2] Alternatively, we can also consider $F$'s with poles at more general positions and then use crossing symmetry to eliminate the dependence on the Mellin amplitude. We consider such functionals in section 4.

All such sum rules have double zeros at the position of double trace operators $\tau = 2\Delta + 2n$, $n \in \mathbb{Z}_{\geq 0}$ above certain twist $\tau_0$ which depends on the choice of $F$ in (2.3). This is due to the simple fact that in (2.4), $\mathcal{Q}_{\ell,m}^{\tau,d}(-2\gamma_{13}) \sim \left(\sin \frac{\tau - 2\Delta}{2}\pi\right)^2$. Sum rules with this property were called "dispersive" in [8].

## 2.1 Polyakov Conditions

The most convenient location for the poles of $F(\gamma_{12}, \gamma_{13})$ is at $\gamma_{ij} = -n$ for $n = 0, 1, 2, \dots$. As discussed in [5], naively, the Mellin amplitude vanishes quadratically at these points. This follows from consistency with the OPE and the assumption that the theory does not contain operators with twists $\tau = 2\Delta + 2n$. The latter condition is not necessary and more generally consistency with the OPE fixes the value of the Mellin amplitude and its first derivative at $\gamma_{ij} = -n$ in terms of the relevant OPE data. This was recently emphasized in [8] and we review it below.

Therefore, in a generic nonperturbative CFT a simple or double pole of $F$ at such points does not contribute to the sum rule (2.3). More precisely, these points are accumulation points of double twist poles of the Mellin amplitude. The double twist operators have twist

$$\tau(n, \ell) = 2\Delta + 2n + \gamma(n, \ell),\qquad(2.5)$$

where $n$ is a non-negative integer and $\gamma(n, \ell) \sim \ell^{-\tau_{gap}}$ for large $\ell$. Let us compute the contribution to the sum rule from the poles accumulating at $\gamma_{12} = -p \in \mathbb{Z}_{\leq 0}$, in the case where $F$ has a pole of order $k$ at this accumulation point,

$$\oint \frac{d\gamma_{12}}{2\pi i} \frac{M(\gamma_{12}, \gamma_{13})}{(\gamma_{12} + p)^k} = \frac{\partial_{\gamma_{12}}^{k-1} M(-p, \gamma_{13})}{\Gamma(k)} + \sum_{n=0}^{p} \sum_{\ell}^{\infty} \frac{1}{(-\gamma(n,\ell)/2)^k} \frac{1}{2} C_{\tau(n,\ell),\ell}^2 \mathcal{Q}_{\ell,p-n}^{\tau(n,\ell),d}(-2\gamma_{13}),\quad(2.6)$$

where the contour integral is taken counterclockwise around $\gamma_{12} = -p$. For $k = 1, 2$ in a generic nonperturbative CFT the first term in the RHS of (2.6) is zero. These are familiar Polyakov conditions. More generally, when there are operators with twist $\tau = 2\Delta + 2p$ in the theory it is non-zero but is again given in terms of the OPE data of the theory (we write the precise formula below). For $k > 2$ we expect $\partial_{\gamma_{12}}^{k-1} M(-p, \gamma_{13})$ to be non-zero and not easily computable given the OPE data of the theory. Therefore below we restrict our analysis to the functionals with $k \leq 2$.

Convergence of this sum is determined by the large spin behaviour of the summand. It is well known that the OPE coefficients approach the ones of mean field theory at large spin [20,21]. Using the asymptotic behaviour (A.21) of the Mack polynomial, we find

$$\frac{1}{(-\gamma(n,\ell)/2)^k} \frac{1}{2} C_{\tau(n,\ell),\ell}^2 \mathcal{Q}_{\ell,p-n}^{\tau(n,\ell),d}(-2\gamma_{13}) \sim \ell^{(k-2)\tau_{gap} - 2\gamma_{13} + 2\Delta - 1} + (\gamma_{13} \leftrightarrow \gamma_{14})\qquad(2.7)$$

where $\gamma_{14} = \Delta + p - \gamma_{13}$. Therefore, the sum over $\ell$ in (2.6) converges if and only if

$$\text{Re}\,\gamma_{13}, \text{Re}\,\gamma_{14} > \Delta - \frac{2-k}{2}\tau_{gap} \quad \Leftrightarrow \quad \Delta - \frac{2-k}{2}\tau_{gap} < \text{Re}\,\gamma_{13} < p + \frac{2-k}{2}\tau_{gap}\qquad(2.8)$$

These convergence regions are shown in figure 1 in red for $k = 1$ and in green for $k = 2$.

The previous analysis can be equivalently stated in terms of the behaviour of the Mellin amplitude close to the accumulation point $\gamma_{12} = -p$,

$$M(\gamma_{12}, \gamma_{13}) \sim (\gamma_{12} + p)^{1 + 2\frac{\gamma_{13} - \Delta}{\tau_{gap}}} + (\gamma_{12} + p)^{1 + 2\frac{\gamma_{14} - \Delta}{\tau_{gap}}} + O(\gamma_{12} + p)^2.\qquad(2.9)$$

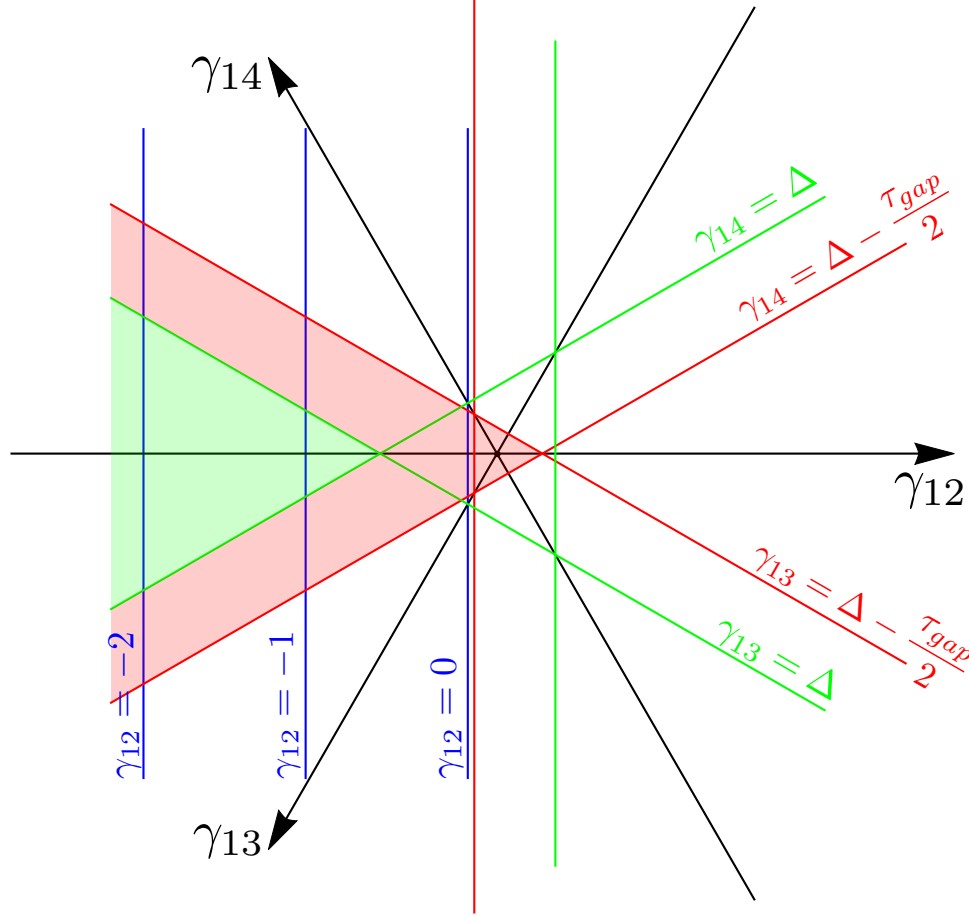

Figure 1: The Mellin-Mandelstam plane. The axis at $120°$ ensure that every point on the plane satisfies $\gamma_{12} + \gamma_{13} + \gamma_{14} = \Delta$. The accumulation points $\gamma_{12} = -n$ for $n = 0, 1, ...$ are shown in blue. The region of convergence for sum rules with a function $F$ with a simple/double pole at one of these points is shown in red/green. Notice that the red region contains the green region.

We conclude that the naive Polyakov conditions

$$M(\gamma_{12} = -p, \gamma_{13}) = 0 \qquad \text{and} \qquad \partial_{\gamma_{12}} M(\gamma_{12} = -p, \gamma_{13}) = 0 \qquad (2.10)$$

can be imposed, respectively, inside the red and green regions of figure 1.

**Exact double twist operators** In writing (2.10) we tacitly assumed that the $s$-channel OPE, or $x_{12}^2 \to 0$, expansion of the correlator does not contain operators of twist $\tau = 2\Delta$. This is the case for the sum rules that we analyze in this paper and in a generic CFT. More generally, we can have

$$M(\gamma_{12} = -p, \gamma_{13}) = a_p(\gamma_{13}) \qquad \text{and} \qquad \partial_{\gamma_{12}} M(\gamma_{12} = -p, \gamma_{13}) = b_p(\gamma_{13}) \qquad (2.11)$$

if the $s$-channel OPE contains the derivative of the conformal block $\partial_\Delta G_{2\Delta+2n+\ell,\ell}(u,v)$ with $n \le p$, which contribute to the former, and conformal blocks $G_{2\Delta+2n+\ell,\ell}(u,v)$ also with $n \le p$, which contribute to the latter. This fact was recently discussed in detail in [8]. Very often a convenient way to derive $a_p(\gamma_{13})$ and $b_p(\gamma_{13})$ is to first use (2.10) and then think of $\partial_\Delta G_{2\Delta+2n+\ell,\ell}(u,v)$ and $G_{2\Delta+2n+\ell,\ell}(u,v)$ as emerging as we expand the nonperturbative sum rules in some small perturbative parameter.

Let us be more precise about it. Imagine we have an operator of twist $\tau = 2\Delta + 2n$ with $n \geq 0$ and spin $\ell$ that appears in the OPE. It is convenient to take $\tau$ slightly away from this value and then take the limit $\tau \to 2\Delta + 2n$. We can compute the behaviour of the factor $M(\gamma_{12}, \gamma_{13})\Gamma^2(\gamma_{12})$ in the integrand in (2.1) close to the pole at $\gamma_{12} = \Delta - \frac{\tau}{2} - (p-n)$ where $p \geq n$. This gives

$$M(\gamma_{12}, \gamma_{13})\Gamma^2(\gamma_{12}) \approx \frac{-\frac{1}{2}C^2_{\tau,\ell}\mathcal{Q}^{\tau,d}_{\ell,p-n}(-2\gamma_{13})}{\gamma_{12} - (\Delta - \frac{\tau}{2} - (p-n))}\Gamma^2\left(\Delta - \frac{\tau}{2} - (p-n)\right)$$

$$\xrightarrow[\tau \to 2\Delta + 2n]{} \frac{-\frac{1}{2}C^2_{2\Delta+2n,\ell}\tilde{\mathcal{Q}}^{2\Delta+2n,d}_{\ell,p-n}(-2\gamma_{13})}{\gamma_{12} + p} , \tag{2.12}$$

where

$$\tilde{\mathcal{Q}}^{2\Delta+2n,d}_{\ell,p}(-2\gamma_{13}) \equiv \lim_{\tau \to 2\Delta + 2n} \mathcal{Q}^{\tau,d}_{\ell,p}(-2\gamma_{13})\Gamma^2\left(\Delta - \frac{\tau}{2} - (p-n)\right). \tag{2.13}$$

This means that an exact double twist operator with $\tau = 2\Delta + 2n$ gives rise to simple poles in the Mellin integrand in (2.1) at $\gamma_{12} = -p$ for all integer $p \geq n$. From this fact together with $\lim_{\gamma_{12} \to -p} \Gamma^2(\gamma_{12})(\gamma_{12} + p)^2 = \frac{1}{p!^2}$ we conclude that the contribution of such an operator to the derivative of the Mellin amplitude $b_p(\gamma_{13})$ takes the form

$$b_p(\gamma_{13}) = -\frac{(p!)^2}{2}C^2_{2\Delta+2n,\ell}\tilde{\mathcal{Q}}^{2\Delta+2n,d}_{\ell,p-n}(-2\gamma_{13}), \qquad p \geq n. \tag{2.14}$$

## 2.2 Family of Functionals

Let us introduce a family of functionals that will be useful in the present paper. They are specified by a simple rational function $F(\gamma_{12}, \gamma_{13})$ that enters into the sum rule (2.3) and takes the following form

$$F_{p_1,p_2,p_3} \equiv \frac{2}{(\gamma_{12} + p_1)(\gamma_{12} + p_2)(\gamma_{14} + p_3)}, \qquad p_i \in \mathbb{Z}_{\geq 0}. \tag{2.15}$$

Note that switching $\gamma_{12} \leftrightarrow \gamma_{14}$ in $F_{p_1,p_2,p_3}$ leads to the same functional due to the crossing symmetry $M(\gamma_{12}, \gamma_{13}) = M(\gamma_{14}, \gamma_{13})$. For fixed $\gamma_{13}$ and large $\gamma_{12}$ we have $F_{p_1,p_2,p_3}(\gamma_{12}, \gamma_{13}) \sim \frac{1}{\gamma_{12}^3}$ and arcs at infinity indeed do not contribute. We denote the corresponging functional $\omega_{p_1,p_2,p_3} \equiv \omega_{F_{p_1,p_2,p_3}}$.

Using the Polyakov conditions (2.10) we get the following sum rule

$$\omega_{p_1,p_2,p_3} = \sum_{\tau > 0, \ell, m = 0}^{\infty} C^2_{\tau,\ell}\omega^{\tau,\ell,m}_{p_1,p_2,p_3} = 0, \tag{2.16}$$

where $\omega^{\tau,\ell,m}_{p_1,p_2,p_3}$ denotes the contribution of a given collinear family of descendants from the primary operator with twist $\tau$ and spin $\ell$ into the sum rule,

$$\omega^{\tau,\ell,m}_{p_1,p_2,p_3} \equiv \frac{\mathcal{Q}^{\tau,d}_{\ell,m}(-2\gamma_{13})}{\prod_{i=1}^{2}(\gamma_{13} - m - p_i - \frac{\tau}{2})(\Delta - m + p_3 - \frac{\tau}{2})} - \frac{\mathcal{Q}^{\tau,d}_{\ell,m}(-2\gamma_{13})}{\prod_{i=1}^{2}(\Delta - m + p_i - \frac{\tau}{2})(-\gamma_{13} + m + p_3 + \frac{\tau}{2})}. \tag{2.17}$$

Let us discuss properties of these functionals. First of all, these functionals are not all independent. Obviously, $\omega_{p_1,p_2,p_3} = \omega_{p_2,p_1,p_3}$. Moreover, it is easy to check that

$$\omega_{p_1,p_2,p_2} = \omega_{p_1,p_2,p_1}. \tag{2.18}$$

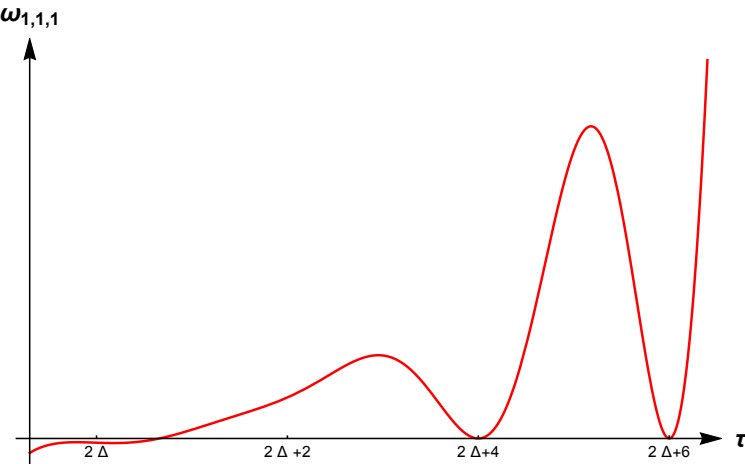

Figure 2: A plot of the $\omega_{1,1,1}^{\tau,\ell} \equiv \sum_{m=0}^{\infty} \omega_{1,1,1}^{\tau,\ell,m}$ functional as a function of the twist $\tau$ of the exchanged operator. We picked $\Delta = \frac{3}{5}$, $\gamma_{13} = \frac{3}{4}$, $\ell = 2$ and $d = 3$. The plot is qualitatively the same for other values. We sum in $m$ from 0 to 50, since this is enough to have an accurate plot of the functional. Notice that the functional does not vanish for twists $\tau = 2\Delta = 1$ and $\tau = 2\Delta + 2 = 3$. However, it has double zeros for all $\tau = 2\Delta + 2n$, for $n \geq 2$.

Next using figure 1 we can understand convergence properties of the functionals $\omega_{p_1,p_2,p_3}$. The dangerous contribution comes from the large spin double twist operators. To discuss this it is convenient to distinguish two cases $p_1 = p_2$ and, without loss of generality, $p_1 > p_2$.

When $p_1 = p_2$ we use the subleading Polyakov condition in $\gamma_{12}$ which converges in the green region in figure 1, we also use the leading Polyakov condition in $\gamma_{14}$ which converges in the crossing transformation of the red region in figure 1. As a result we conclude that

$$\omega_{p,p,p_3} \text{ converges when } p \geq 1, \; p_3 > 0, \quad \gamma_{13} \in \text{green region} \quad \Leftrightarrow \quad \Delta < \operatorname{Re}\gamma_{13} < \min\left(p, p_3 + \frac{\tau_{gap}}{2}\right) \tag{2.19}$$

When $p_1 > p_2$ we use the leading Polyakov condition both in $\gamma_{12}$ and $\gamma_{14}$ which converges in the red region (and its crossing transformation) in figure 1 so no extra constraints arise and we have

$$\omega_{p_1,p_2,p_3} \text{ converges when } \quad p_1 > p_2, \quad \gamma_{13} \in \text{red region}, \quad \Leftrightarrow \quad \Delta - \frac{\tau_{gap}}{2} < \operatorname{Re}\gamma_{13} < \min(p_2, p_3) + \frac{\tau_{gap}}{2} \tag{2.20}$$

The defining property of $\omega_{p_1,p_2,p_3}$ functional is sensitivity to the double twist operators with twists $\tau = 2\Delta + 2n$, where $n \leq p_i$. Indeed, the Mack polynomials $\mathcal{Q}_{\ell,m}^{\tau,d}(-2\gamma_{13})$ have double zeros at $\tau = 2\Delta + 2n$, whereas the brackets in (2.16) have poles at $\tau = 2\Delta + 2(p_i - m)$ which enhances the contribution of the corresponding double twist families. See figures 2 and 3.

In our analysis of the Wilson-Fisher model in $d = 4 - \epsilon$ dimensions below we will only study the properties of the leading, $n = 0$, and the first sub-leading, $n = 1$, double twist family of operators. This naturally restrict our attention to the functionals with $p_i \leq 1$. Together with linear dependence and convergence properties explained above it leaves us with two functionals

$$\omega_{1,0,0}, \quad \omega_{1,1,1}. \tag{2.21}$$

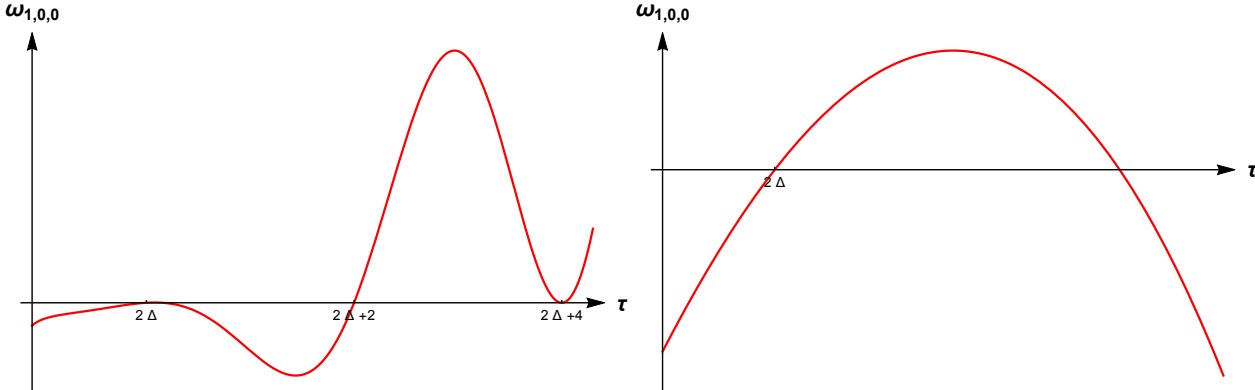

Figure 3: On the left, a plot of the $\omega_{1,0,0}$ functional as a function of the twist $\tau$ of the exchanged operator. We picked $\Delta = 1.1$, $\gamma_{13} = \frac{1}{3}$, $\ell = 2$ and $d = 4$. The plot is qualitatively similar for other values. We sum in $m$ from 0 to 50, since this is enough to have an accurate plot of the functional. Notice that the functional has single zeros for twists $\tau = 2\Delta = 2$ (though it is not very clear from the left plot) and $\tau = 2\Delta + 2 = 4$. However, it has double zeros for all $\tau = 2\Delta + 2n$, where $n \geq 2$. On the right, we zoom in to the region around $\tau = 2\Delta$, so that we can observe the single zero at $\tau = 2\Delta$.

## 2.3 Check with Mean Field Theory

In mean field theory (MFT), there are only exact double twist operators in the OPE $\phi \times \phi$, where $\phi$ is the gaussian field. Therefore, the functionals $\omega_{p_1,p_2,p_3}$ annihilate every term of the sum in (2.16) because the functions $\mathcal{Q}_{\ell,m}^{\tau,d}(-2\gamma_{13})$ have double zeros at $\tau - 2\Delta \in \mathbb{Z}_{\geq 0}$. This conclusion is not correct if we choose $p_1 = p_2 \in \mathbb{Z}_{\geq 0}$ because the double pole of $F$ at $\gamma_{12} = p_1$ cancels the double zero.

For concreteness consider the functional $\omega_{1,1,p}$. Using the MFT OPE coefficients, we obtain

$$\omega_{1,1,p} = -\frac{2^{2+2\Delta}}{\sqrt{\pi}\,\Gamma^2(\Delta)(1 + \Delta - \gamma_{13} + p)} \sum_{\substack{\ell=0 \\ even}}^{\infty} \frac{2^\ell \Gamma\left(\frac{1}{2} + \Delta + \ell\right) P_\ell(\gamma_{13})}{(d + 2\ell)(2 - d + 4\Delta + 2\ell)\ell!\,\Gamma(\Delta + \ell + 1)}\,, \qquad (2.22)$$

where

$$P_\ell(\gamma_{13}) = (d + 2\ell)(\ell + \Delta)Q_{\ell,1}^{\tau=2\Delta,d}(-2\gamma_{13}) + (2 - d + 2\Delta)(2\Delta + 2\ell + 1)Q_{\ell,0}^{\tau=2\Delta+2,d}(-2\gamma_{13})\,. \quad (2.23)$$

Here we used the (non-calligraphic) Mack polynomials $Q_{\ell,m}^{\tau,d}$ defined in appendix A. Notice that only the double-twist operators with twist $\tau = 2\Delta$ and $\tau = 2\Delta + 2$ contribute to this sum rule. The polynomial $P_\ell(\gamma_{13})$ inherits the symmetry $P_\ell(\gamma_{13}) = P_\ell(1 + \Delta - \gamma_{13})$ from the Mack polynomials.

As predicted by equation (2.7), the large $\ell$ behaviour of the summand in (2.22) is given by

$$\sim \ell^{\max(2\Delta - 2\gamma_{13} - 1, 2\gamma_{13} - 3)}\,, \qquad (2.24)$$

which implies convergence of the sum over $\ell$ for $\Delta < \gamma_{13} < 1$. In figure 4, we plot the partial sums

$$S_J(\gamma_{13}) = \sum_{\substack{\ell=0 \\ even}}^{J} \frac{2^\ell \Gamma\left(\frac{1}{2} + \Delta + \ell\right) P_\ell(\gamma_{13})}{(d + 2\ell)(2 - d + 4\Delta + 2\ell)\ell!\,\Gamma(\Delta + \ell + 1)}\,, \qquad (2.25)$$

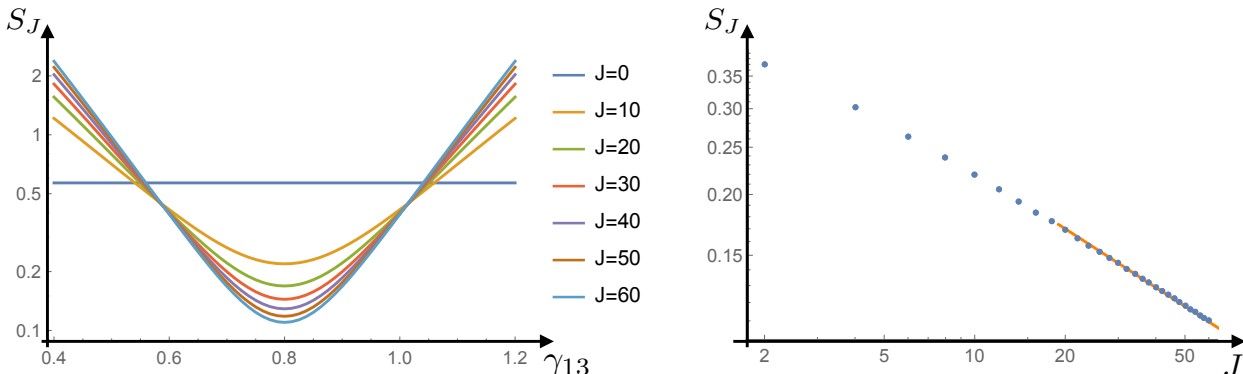

Figure 4: Partial sum $S_J(\gamma_{13})$ defined in (2.25) for $d = 3$ and $\Delta = \frac{3}{5}$. On the left, one can see that only for $\Delta < \gamma_{13} < 1$ the partial sum converges to zero as expected. On the right, we fix $\gamma_{13} = \frac{4}{5}$ and use a log-log plot to exhibit the large $J$ behavior predicted by (2.26). The straight orange line is a fit (to the points $20 \le J \le 60$) with slope given by (2.26).

for several values of $J$. One can see that $S_J(\gamma_{13})$ tends to zero when $J \to \infty$ if $\Delta < \gamma_{13} < 1$ and it diverges otherwise. Moreover, one can also check the large $J$ behaviour

$$\log S_J(\gamma_{13}) \approx \max\left(2\Delta - 2\gamma_{13}, 2\gamma_{13} - 2\right) \log J\,, \tag{2.26}$$

in agreement with (2.24).

The knowledgeable reader may ask: how can we get a sum rule for MFT using Mellin amplitudes? Indeed, the Mellin amplitude for MFT vanishes identically [5]. One way to understand the success of the exercise above is as follows. Consider an interacting theory with a continuous coupling $\lambda$, such that at $\lambda = 0$ we obtain MFT. The Mellin amplitude is non-trivial and leads to the sum rules (2.16) for any $\lambda > 0$. Then, one obtains the sum rule (2.22) for MFT in the limit $\lambda \to 0$.

Let us now consider the functional

$$\omega_{0,0,p} = -\frac{2^{1+2\Delta}}{\sqrt{\pi}\,\Gamma^2(\Delta)(\Delta - \gamma_{13} + p)} \sum_{\substack{\ell=0 \\ even}}^{\infty} \frac{2^{\ell}\Gamma\left(\frac{1}{2} + \Delta + \ell\right)}{\ell!\,\Gamma(\Delta + \ell)} Q_{\ell,0}^{\tau=2\Delta,d}(-2\gamma_{13})\,. \tag{2.27}$$

One can check that this sum vanishes for $\Delta < \mathrm{Re}\,\gamma_{13} < 0$ in agreement with the general formula (2.19). Notice that in unitary CFTs there is no convergence region for this sum rule because $\Delta < 0$ is forbidden. Nevertheless, we shall use the functional

$$\omega_{0,0,0}$$

in the $\epsilon$–expansion by applying it to the difference between the CFT data in the interacting theory and in MFT. This trick will give rise to a finite region of convergence for the sum rule $\omega_{0,0,0}$.

## 3 $\epsilon$ − expansion

In this section we apply the dispersive Mellin sum rules described in the previous section to the Wilson-Fisher fixed point in $d = 4 - \epsilon$ spacetime dimensions. This theory contains the lightest scalar operator $\phi$ of dimension $\Delta_\phi$ and we consider the four-point $\langle\phi\phi\phi\phi\rangle$ and the associated CFT data which includes the following operators:

- The lightest scalar that appears in the OPE $\phi \times \phi$. We denote this operator by $\phi^2$ with dimension $\Delta_{\phi^2}$ and OPE coefficient $C_{\phi^2}$.

- The leading twist operators $j_\ell$ (also called twist-two) with twist $\tau_\ell$ and the three-point function $C_{j_\ell}$. These are non-degenerate (there is a single operator for every even spin $\ell \geq 2$) and can be identified with the double twist operators with $n = 0$.

- Twist-four operators of even spin $\ell \geq 0$. These are non-degenerate for $\ell = 0, 2$ [22], and degenerate for $\ell \geq 4$. They also include the double twist family with $n = 1$. We will study their OPE data on average.

- Higher twist operators. These have twist six and higher when $\epsilon = 0$. We do not say anything about these operators. They will not appear in the dispersive Mellin sum rules to the perturbative order that we analyze them.

From the bootstrap point of view this model can be defined as follows. We start with $\langle \phi\phi\phi\phi \rangle$ being the mean field theory correlator in $d = 4 - \epsilon$ dimensions. Next we assume that the CFT data (scaling dimensions and OPE coefficients) depends on $\epsilon$ as a power series. We then study the crossing equations perturbatively in $\epsilon$. Such perturbative solutions to crossing were analyzed in [23] and they include infinitely many ambiguities due to contact interactions in $AdS$. It is reasonable to conjecture that these are completely fixed by requiring that at every order in $\epsilon$ the correlator satisfies the Regge bound (2.2) and that the stress energy tensor is conserved. In particular, this means that we can use the dispersive Mellin sum rules order by order in $\epsilon$.

## 3.1 Review and Notation

Let us state our definitions. The known results for the OPE data in the Wilson-Fisher fixed point in $d = 4 - \epsilon$ and the relevant references can be found in appendix B.

We follow [17] and define the expansion parameter $g$ to be the anomalous dimension of $\phi^2$,

$$\Delta_{\phi^2} = 2\Delta_\phi + g, \tag{3.1}$$

Then, the spacetime dimensionality $d$

$$d = 4 + a_1 g + a_2 g^2 + a_3 g^3 + a_4 g^4 + \dots. \tag{3.2}$$

Using this equation one can find $g$ as a function of $\epsilon$ and vice versa. For the conformal dimension of $\phi$ we write

$$\Delta_\phi = \frac{d-2}{2} + \gamma_1(\phi)g + \gamma_2(\phi)g^2 + \gamma_3(\phi)g^3 + \gamma_4(\phi)g^4 + \dots. \tag{3.3}$$

We will derive that $\gamma_1(\phi) = 0$. So, the correction to the dimension of $\phi$ starts at order $g^2$. Similarly, we write for the twist of the twist-two operators

$$\tau_\ell = 2\Delta_\phi + \gamma_1(j_\ell)g + \gamma_2(j_\ell)g^2 + \gamma_3(j_\ell)g^3 + \gamma_4(j_\ell)g^4 + \dots, \quad \ell \geq 2. \tag{3.4}$$

We will derive that $\gamma_1(j_\ell) = 0$. The twist of the stress energy tensor is protected $\tau_2 = d - 2$. For the twist-four operators due to the degeneracy we only compute the averaged values of the relevant OPE data

$$\tau_{4,i}(\ell)) = 2\Delta_\phi + 2 + \gamma_1(\tau_{4,i}(\ell))g + \dots, \tag{3.5}$$

where $i$ denotes various degenerate operators at $g = 0$. We will only compute averaged moments of the twist-four anomalous dimensions as follows

$$\langle \gamma(\tau_4(\ell))^n \rangle \equiv \frac{\sum_i C^2_{\tau_{4,i}(\ell)} \gamma^n(\tau_{4,i}(\ell))}{\sum_i C^2_{\tau_{4,i}(\ell)}}, \tag{3.6}$$

where the sum is over the degenerate operators. Note that $\ell = 0, 2$ operators are not degenerate.

Concerning OPE coefficients, we define them relative to the mean field theory. In MFT, the square of the OPE coefficient of the operator of dimension $2\Delta_\phi + 2n + \ell$ and spin $\ell$ is equal to

$$C^2_{n,\ell} = \frac{\left(1 + (-1)^\ell\right)\left(\Delta - \frac{d}{2} + 1\right)_n^2 (\Delta)^2_{\ell+n}}{\Gamma(\ell+1)\Gamma(n+1)\left(\frac{d}{2} + \ell\right)_n (2\Delta - d + n + 1)_n (2\Delta + \ell + 2n - 1)_\ell \left(2\Delta - \frac{d}{2} + \ell + n\right)_\ell} \tag{3.7}$$

where $\Delta_\phi$ is the dimension of the fundamental field and $d$ is the spacetime dimension, both of which are nontrivial functions of $g$.

We parametrise the OPE coefficients of $\phi^2$, of the leading twist operators $j_\ell$ and the sum of squares of OPE coefficients over degenerate twist 4 operators $\tau_4(l)$ as

$$C^2_{\phi^2} = C^2_{0,0} \times \left(1 + c_1(\phi^2)g + c_2(\phi^2)g^2 + c_3(\phi^2)g^3 + c_4(\phi^2)g^4 + ...\right),$$

$$C^2_{j_\ell} = C^2_{0,\ell} \times \left(1 + c_1(\ell)g + c_2(j_\ell)g^2 + c_3(j_\ell)g^3 + c_4(j_\ell)g^4 + ...\right), \tag{3.8}$$

$$\sum_i C^2_{\tau_{4,i}(\ell)} = C^2_{1,\ell} \times \left(b_2(\ell) + b_3(\ell)g + ...\right) \sim O(g^2),$$

respectively, where we plug $\Delta_\phi$ and $d$ as functions of $g$ as above. In the last line we emphasized that since in the free field theory $C^2_{n \geq 1, \ell}|_{\Delta = \frac{d-2}{2}} = 0$ the twist four operators, or $n = 1$, first contribute at order $g^2$. Higher twist operators, or $n \geq 2$, first contribute at order $g^4$.

The quantities $c_i(\phi^2)$, $c_i(j_\ell)$, and $b_i(\ell)$ are to be determined using the dispersive Mellin functionals below.

## 3.2 Order $g^0$

Let us discuss the action of the functionals $\omega_{1,0,0}$, $\omega_{1,1,1}$ and $\omega_{0,0,0}$ at order $g^0$. When $g = 0$ the correlator $\langle \phi\phi\phi\phi \rangle$ is the one of the free scalar field in 4 dimensions. The relevant action of the functionals on mean field theory was discussed in section 2.3 to which we refer the reader for the relevant formulas. In the following applications, we always consider the difference between the action of the functionals on the Wilson-Fischer fixed point and their action on mean field theory.

## 3.3 Order $g^1$

Let us discuss the action of $\omega_{1,0,0}$ and $\omega_{0,0,0}$. $\omega_{1,0,0}$ has single zeros for the leading and subleading twist trajectories, whereas it has double zeros for the other twist trajectories. For this reason, the contribution of the twist 2 and twist 4 operators comes proportional to their anomalous dimensions, whereas the contribution of twist 6 or higher operators comes proportional to their anomalous dimensions squared. For this reason, twist 6 operators do not contribute at this order. Furthermore, the OPE coefficients of twist 4 operators vanish at order $g^0$. Thus, twist 4 operators do not contribute to $\omega_{1,0,0}$ to first order in $g$.

So, only twist 2 operators contribute at first order in $g$. Their contribution is given by

$$\omega_{1,0,0}|_{g^1} = \frac{2}{(-2+\gamma_{13})}\left(Q^{\tau=2,d=4}_{\ell=0,m=0}(-2\gamma_{13}) - Q^{\tau=2,d=4}_{\ell=0,m=1}(-2\gamma_{13})\right) \tag{3.9}$$

$$+ \sum_{\substack{\ell=2 \\ \text{even}}}^{\infty} \frac{2^{\ell+2}\Gamma(\ell+\frac{3}{2})}{(-2+\gamma_{13})\sqrt{\pi}\Gamma(\ell+1)^2}\left(Q^{\tau=2,d=4}_{\ell,m=0}(-2\gamma_{13}) - Q^{\tau=2,d=4}_{\ell,m=1}(-2\gamma_{13})\right)\gamma_1(j_\ell)$$

The first line corresponds to the contribution of $\phi^2$ and the second line corresponds to the contribution of the operators in the leading twist trajectory with even spin $\ell \geq 2$. It turns out that the contribution of $\phi^2$ vanishes identically since $Q^{\tau=2,d=4}_{\ell=0,m=0}(-2\gamma_{13}) = Q^{\tau=2,d=4}_{\ell=0,m=1}(-2\gamma_{13}) = 1$.

We will want to apply the orthogonality relation (A.8) to (3.9). The decomposition

$$Q^{\tau=2,d=4}_{\ell_1,m=1}(s) = Q^{\tau=2,d=4}_{\ell_1,m=0}(s) + \sum_{\ell_2=0}^{\ell_1-1} \frac{2^{-\ell_1+\ell_2+1}\Gamma(\ell_1)\Gamma(\ell_1+1)\Gamma\left(\ell_2+\frac{3}{2}\right)}{\Gamma\left(\ell_1+\frac{1}{2}\right)\Gamma(\ell_2+1)^2}Q^{\tau=2,d=4}_{\ell_2,m=0}(s) \tag{3.10}$$

will be important. Furthermore, let us define

$$\zeta_1 = \sum_{\substack{\ell=2 \\ \text{even}}}^{\infty} \frac{\ell+\frac{1}{2}}{\ell}\gamma_1(j_\ell). \tag{3.11}$$

Then, by permuting the order of the sums, (3.9) can be rewritten as

$$(-2+\gamma_{13})\omega_{1,0,0}|_{g^1} = -\sum_{\ell=0}^{\infty} \frac{2^{\ell+3}}{\sqrt{\pi}}\frac{\Gamma(\ell+\frac{3}{2})}{\Gamma(\ell+1)^2}\left(\zeta_1 - \sum_{\substack{\ell_1=2 \\ \text{even}}}^{\ell} \frac{\ell_1+\frac{1}{2}}{\ell_1}\gamma_1(j_{\ell_1})\right)Q^{\tau=2,d=4}_{\ell,m=0}(-2\gamma_{13}). \tag{3.12}$$

Since the Mack polynomials are orthogonal (see (A.8)), every term in the sum over $\ell$ in (3.12) must vanish. Then, the term with $\ell=0$ implies that $\zeta_1=0$. The term with $\ell=2$ implies $\gamma_1(j_{\ell=2})=0$. The term with $\ell=4$ implies $\gamma_1(j_{\ell=4})=0$ and so on. We therefore conclude that

$$\boxed{\gamma_1(j_\ell)=0} \tag{3.13}$$

Conservation of the stress tensor implies that $2\Delta_\phi + \gamma(j_{\ell=2}) = d-2$. Expanding this to first order in $g$ and using $\gamma_1(j_{\ell=2})=0$, we obtain

$$\boxed{\gamma_1(\phi)=0} \tag{3.14}$$

Let us consider now the action of $\omega_{0,0,0}$. This functional does not vanish for the leading twist trajectory, but it has double zeros for all the subleading twist trajectories. For this reason only the twist 2 operators contribute to first order in $g$. We have that

$$\omega_{1,0,0}|_{g^1} = -\sum_{\substack{\ell=2 \\ \text{even}}}^{\infty} \frac{2^{\ell+3}\Gamma(\ell+\frac{3}{2})}{(-1+\gamma_{13})\sqrt{\pi}\Gamma(\ell+1)^2}c_1(\ell)Q^{\tau=2,d=4}_{\ell,m=0}(-2\gamma_{13}) - \frac{4(1+c_1(\phi^2))}{(-1+\gamma_{13})}Q^{\tau=2,d=4}_{\ell=0,m=0}(-2\gamma_{13}). \tag{3.15}$$

The orthogonality relation (A.8) applied to $(-1+\gamma_{13})\times$(3.15) immediately implies that

$$\boxed{c_1(\phi^2)=-1, \quad c_1(\ell)=0} \tag{3.16}$$

## 3.4 Order $g^2$

Let us consider the action of the $\omega_{1,0,0}$ functional. At order $g^2$, only $\phi^2$ and operators in the leading Regge trajectory $j_\ell$ contribute. The twist four and higher operators do not contribute, since their OPE coefficients start at order $g^2$ and the functional $\omega_{1,0,0}$ vanishes for the exchange of sub-leading twists.

The contribution of $\phi^2$ is equal to

$$\sum_{m=0}^{\infty} C_{\phi^2}^2 \omega_{1,0,0}^{\Delta_{\phi^2},0,m}\bigg|_{g^2} = \frac{-2 + a_1(-1 + \gamma_{13}) + \gamma_{13}}{(-2 + \gamma_{13})(-1 + \gamma_{13})}. \tag{3.17}$$

The contribution of the leading twist trajectory is equal to

$$\sum_{\substack{\ell=2 \\ \text{even}}}^{\infty} C_{j_\ell}^2 \omega_{1,0,0}^{\tau_\ell,\ell,0}|_{g^2} = \sum_{\substack{\ell=2 \\ \text{even}}}^{\infty} \frac{2^{\ell+2}\Gamma(\ell+\frac{3}{2})\big((\ell+1)Q_{\ell,m=0}^{\tau=2,d=4}(-2\gamma_{13}) - Q_{\ell,m=1}^{\tau=2,d=4}(-2\gamma_{13})\big)\gamma_2(j_\ell)}{(-2 + \gamma_{13})(\ell+1)\sqrt{\pi}\Gamma(\ell+1)^2}. \tag{3.18}$$

Recall that $\gamma_2(j_\ell)$ is the anomalous dimension of the leading twist operators.

In order to extract the anomalous dimensions $\gamma_2(j_\ell)$ we will use the orthogonality relation (A.8) among $m = 0$ Mack polynomials. We will also use equation (3.10) to decompose $m = 1$ Mack polynomials into $m = 0$ Mack polynomials. From the sum rule (3.17) + (3.18) = 0 we can obtain several equations by multiplying it by $(-2 + \gamma_{13})\Gamma^2(\gamma_{13})\Gamma^2(1 - \gamma_{13})Q_{\ell,m=0}^{\tau=2,d=4}(-2\gamma_{13})$, integrating over $\gamma_{13}$ and using the orthogonality relation (A.8). For even $\ell$ we obtain

$$-\frac{\sqrt{\pi}2^{1-\ell}\ell\gamma_2(j_\ell)\Gamma(\ell+1)^2}{(\ell+1)\Gamma\left(\ell+\frac{1}{2}\right)} + \frac{\sqrt{\pi}2^{1-\ell}\Gamma(\ell+1)^2\left(\zeta - \sum_{\ell_1=2}^{\ell}\left[\frac{(2\ell_1+1)\gamma_2(j_{\ell_1})}{\ell_1(\ell_1+1)}\right]\right)}{\Gamma\left(\ell+\frac{1}{2}\right)}$$

$$= \int_{-1-i\infty}^{-1+i\infty} \frac{ds}{4\pi i}\left(a_1 + \frac{4+s}{2+s}\right)\Gamma\left(-\frac{s}{2}\right)^2\Gamma\left(\frac{s}{2}+1\right)^2 Q_{\ell,m=0}^{\tau=2,d=4}(s) \tag{3.19}$$

$$= -\frac{\sqrt{\pi}2^{-\ell-1}\Gamma(\ell+1)^2\left((\ell+1)^2\psi^{(1)}\left(\frac{\ell}{2}+1\right) - (\ell+1)^2\psi^{(1)}\left(\frac{\ell+3}{2}\right) - 4\right)}{(\ell+1)^2\Gamma\left(\ell+\frac{1}{2}\right)} + \delta_{\ell,0}(1+a_1),$$

where $s = -2\gamma_{13}$, $\psi^{(1)}(x)$ is the order 1 polygamma function and we compute the integral above in appendix C.2. We also introduced the quantity

$$\zeta \equiv \sum_{\substack{\ell=2 \\ \text{even}}}^{\infty} \frac{(2\ell+1)}{\ell(\ell+1)}\gamma_2(j_\ell). \tag{3.20}$$

We treat $\zeta$ as an independent parameter from the anomalous dimensions, that we will compute.

Furthermore, the orthogonality relation with respect to odd spin Mack polynomials is also very useful. It is given by

$$-\frac{\sqrt{\pi}2^{1-\ell}\Gamma(\ell+1)^2\left(\zeta - \sum_{\ell_1=2}^{\ell-1}\left[\frac{(2\ell_1+1)\gamma_2(j_{\ell_1})}{\ell_1(\ell_1+1)}\right]\right)}{\Gamma\left(\ell+\frac{1}{2}\right)} \tag{3.21}$$

$$= -\frac{\sqrt{\pi}2^{-\ell-1}\Gamma(\ell+1)^2\left((\ell+1)^2\psi^{(1)}\left(\frac{\ell}{2}+1\right) - (\ell+1)^2\psi^{(1)}\left(\frac{\ell+3}{2}\right) - 4\right)}{(\ell+1)^2\Gamma\left(\ell+\frac{1}{2}\right)}. \tag{3.22}$$

We used the spin 1 equation to determine $\zeta = \frac{\pi^2}{12} - 1$.

Knowing $\zeta$, then equations (3.19) can be solved in the following manner. The spin 0 and spin 2 equations determine $a_1$ and $\gamma_2(2)$. The spin 4 equation determines $\gamma_2(4)$ and so on. More generically,

$$a_1 = -3, \quad \gamma_2(j_\ell) = -\frac{1}{\ell(\ell+1)} \tag{3.23}$$

This agrees with known results.

Since we already know the anomalous dimensions of the leading twist operators, we can fix the dimension of $\phi$ by demanding that the stress tensor has twist $d - 2$

$$2\gamma_2(\phi) + \gamma_2(2) = 0 \;\Rightarrow\; \gamma_2(\phi) = \frac{1}{12} \tag{3.24}$$

In order to compute the corrections to the OPE coefficients of the operators in the leading twist trajectory, let us consider the action of $\omega_{0,0,0}$. At order $g^2$ only $\phi^2$ and the leading twist trajectory contribute. The contribution of $\phi^2$ is

$$\sum_{m=0}^{\infty} C_{\phi^2}^2 \omega_{0,0,0}^{\Delta_{\phi^2},0,m}|_{g^2} = 2\left(\frac{2a_1}{\gamma_{13}-1} - \frac{2c_2(\phi^2)}{\gamma_{13}-1} + \frac{4(\gamma_{13}-2)\gamma_{13} + (\gamma_{13}-1)^2\psi^{(1)}(2-\gamma_{13}) + 5}{2(\gamma_{13}-1)^3}\right), \tag{3.25}$$

The contribution of the leading twist trajectory is

$$\sum_{\substack{\ell=2 \\ \text{even}}}^{\infty} C_{j_\ell}^2 \omega_{0,0,0}^{\tau_\ell,\ell,0}|_{g^2} = -2\sum_{\substack{\ell=2 \\ \text{even}}}^{\infty} \frac{2^{\ell+2}\Gamma\left(\ell+\frac{3}{2}\right)}{\sqrt{\pi}(\gamma_{13}-1)\Gamma(\ell+1)^2} \tag{3.26}$$

$$\times \left(Q_{\ell,m=0}^{\tau=2,d=4}(-2\gamma_{13})\left(c_2(j_\ell) + \gamma_2(j_\ell)\left(2S_1(2\ell) - 3S_1(\ell) + \frac{1}{2(\ell+\frac{1}{2})}\right)\right) + \gamma_2(j_\ell)\frac{d}{d\tau}Q_{\ell,m=0}^{\tau=2,d=4}(-2\gamma_{13})\right),$$

The above series converges for $0 < Re(\gamma_{13}) < \frac{1}{2}$. We determined $c_2(\ell)$ and $c_2(\phi^2)$ in the following manner. The series (3.26) contains a part proportional to $c_2(\ell)$ and another part proportional to $\gamma_2(\ell)$. We have already determined $\gamma_2(\ell)$ in the preceding section. So, we can sum the part of the series (3.26) that is proportional to $\gamma_2(\ell)$. In practice, we used, from spin 2 to 100, the exact expressions for Mack polynomials and, from spin 102 to infinity, we used the approximation (A.21), (A.22) to Mack polynomials to sum the tails.

After doing this summation we can apply the orthogonality relation (A.8). We evaluate such integrals numerically. Proceeding in this manner we obtained

$$c_2(\phi^2) = -1 \tag{3.27}$$

We also obtained $c_2(j_\ell)$ for low spins. It precisely matches the formula[3]

$$c_2(j_\ell) = \frac{S_1(2\ell) - S_1(\ell) + \frac{1}{\ell+1}}{\ell(\ell+1)} \tag{3.28}$$

---

[3]At low orders in $g$, our work consists in rederiving known formulas, so we do not keep track of error bars when executing our numerical procedure. However, when making new predictions, we were careful with errors and we do present our results with error bars.

derived in [14], where

$$S_n(\ell) = \sum_{m=1}^{\ell} \frac{1}{m^n}.$$ (3.29)

Thus far we have deduced the conformal data of $\phi$, $\phi^2$ and the leading twist trajectory to order $g^2$. Let us compute the OPE coefficients of the twist 4 operators at order $g^2$, using the $\omega_{1,1,1}$ functional. The contribution of $\phi^2$ is equal to

$$\sum_{m=0}^{\infty} C_{\phi^2}^2 \omega_{1,1,1}^{\Delta_{\phi^2},0,m}\big|_{g^2} = 2\Big(\frac{a_1}{\gamma_{13} - 3} - \frac{2c_2(\phi^2)}{\gamma_{13} - 3} + \frac{2(\gamma_{13} - 5)\gamma_{13}((\gamma_{13} - 5)\gamma_{13} + 13) + 85}{2(\gamma_{13} - 3)^3(\gamma_{13} - 2)^2} + \frac{\psi^{(1)}(4 - \gamma_{13})}{2(\gamma_{13} - 3)}\Big).$$ (3.30)

Of course we already know the values of $a_1$ and $c_2(\phi^2)$, we are just exhibiting which CFT data matters for the action of $\omega_{1,1,1}$.

The contribution of the leading twist trajectory is given by

$$\sum_{\substack{\ell=2 \\ \text{even}}}^{\infty} C_{j_\ell}^2 \omega_{1,1,1}^{\tau_\ell,\ell,0}\big|_{g^2} = -2 \sum_{\substack{\ell=2 \\ \text{even}}}^{\infty} \frac{2^{\ell+2}\Gamma(\ell + \frac{3}{2})}{\sqrt{\pi}(\gamma_{13} - 3)\Gamma(\ell + 1)\Gamma(\ell + 2)}$$ (3.31)

$$\times \Big( Q_{\ell,m=1}^{\tau=2,d=4}(-2\gamma_{13})\Big(c_2(j_\ell) + \gamma_2(j_\ell)(2S_1(2\ell) - 3S_1(\ell) + \frac{1}{2(\ell + \frac{1}{2})} - \frac{1}{\ell + 1} + 1)\Big)$$

$$+ \gamma_2(j_\ell)\frac{d}{d\tau}Q_{\ell,m=1}^{\tau=2,d=4}(-2\gamma_{13})\Big)$$

The contribution of the subleading twist trajectory is equal to

$$\sum_{\substack{\ell=0 \\ \text{even}}}^{\infty} \sum_i C_{\tau_{4,i}(\ell)}^2 \omega_{1,1,1}^{\tau_{4,i}(\ell),\ell,0}\big|_{g^2} = \sum_{\substack{\ell=0 \\ \text{even}}}^{\infty} \Big( -\frac{2\gamma_2(\phi)2^{\ell+3}(b_2(\ell) - 1)\Gamma\left(\ell + \frac{5}{2}\right) Q_{\ell,m=0}^{\tau=4,d=4}(-2\gamma_{13})}{\sqrt{\pi}(\gamma_{13} - 3)(\ell + 1)(\ell + 2)\Gamma(\ell + 1)\Gamma(\ell + 2)}\Big)g^2. $$ (3.32)

Our goal is to compute the averaged OPE coefficients of the twist 4 operators, which we call $b_2(\ell)$. For this reason we will use the orthogonality relation with respect to $m = 0$ twist 4 Mack polynomials. (3.31) is a complicated expression, however we can in principle evaluate it, since we already know all of the conformal data that enters the expression. This involves summing tails, like before. After doing such a summation and using the orthogonality relation (numerically), we managed to obtain $b_2(\ell)$ at low spins. Our results agree with the analytical expression

$$\boxed{b_2(\ell) = 1 + \frac{6}{(\ell + 1)(\ell + 2)}}$$ (3.33)

derived in [17].

## 3.5  Order $g^3$

Let us compute the order $g^3$ corrections to the OPE coefficients of the leading twist trajectory. We assume the formulas for $a_2$ and $\gamma_3(j_\ell)$ [17], see appendix B.

We will use the $\omega_{0,0,0}$ functional. The numerical procedure to extract CFT data is the same as before. The action of the $\omega_{0,0,0}$ functional allows us to calculate the OPE coefficients $c_3(\phi^2)$ and $c_3(j_l)$ at low spins. This precisely matches the analytic formulas [15]

$$c_3(j_\ell) = \frac{3(S_1(\ell) - S_1(2\ell) + S_2(2\ell)) - 2\big(S_1^2(\ell) - S_1(\ell)S_1(2\ell) + S_2(2\ell)\big)}{\ell(\ell+1)} + \frac{3(\ell + \frac{1}{2})(S_1(2\ell) - \frac{\ell-1}{\ell+1}) - (\ell + \frac{3}{2})S_1(\ell)}{\ell^2(\ell+1)^2}$$

(3.34)

$$c_3(\phi^2) = \frac{5}{6} + \frac{7\zeta(3)}{4}$$

(3.35)

Let us turn our attention to the twist 4 operators. Let us apply the $\omega_{1,0,0}$ functional, in order to obtain their first order correction to the anomalous dimensions. Since their OPE coefficients start at order $g^2$, this means we need to study the $\omega_{1,0,0}$ functional to order $g^3$.

There are three types of contributions: from $\phi^2$, from the leading twist operators and from the subleading twist operators. The contribution of $\phi^2$ is equal to

$$\sum_{m=0}^{\infty} C_{\phi^2}^2 \omega_{1,0,0}^{\Delta_{\phi^2},0,m}|_{g^3} = -\frac{3S_1(-\gamma_{13})}{2\left(\gamma_{13}^2 - 3\gamma_{13} + 2\right)} + \frac{\gamma_E(6\gamma_{13} - 3)}{\gamma_{13}^2 - 3\gamma_{13} + 2} + \frac{2\gamma_{13}(6(\gamma_{13} - 3)\gamma_{13} + 19) - 19}{2\left(\gamma_{13}^2 - 3\gamma_{13} + 2\right)^2}$$

where we used the Euler Gamma constant $\gamma_E$.

The contribution of the leading twist operators is equal to

$$\sum_{\substack{\ell=2 \\ \text{even}}}^{\infty} C_{j_\ell}^2 \omega_{1,0,0}^{\tau_\ell,\ell,0}|_{g^3} = \sum_{\substack{\ell=2 \\ \text{even}}} \left( \frac{2^\ell \Gamma(\ell + \frac{3}{2})}{\sqrt{\pi}(\gamma_{13} - 2)^2 \ell^2 \Gamma(\ell+2)^2} \times \left(r_1(\ell) Q_{\ell,m=0}^{\tau=2,d=4}(-2\gamma_{13}) + r_2(\ell) Q_{\ell,m=1}^{\tau=2,d=4}(-2\gamma_{13})\right)\right.$$

(3.36)

$$\left. + \frac{3\,2^{\ell+3}\Gamma\left(\ell + \frac{3}{2}\right)}{\sqrt{\pi}(4 - 2\gamma_{13})\ell\Gamma(\ell+2)^2}\left(\partial_d Q_{\ell,m=1}^{\tau=2,d=4}(-2\gamma_{13}) - (1+\ell)\partial_\tau Q_{\ell,m=0}^{\tau=2,d=4}(-2\gamma_{13}) + \partial_\tau Q_{\ell,m=1}^{\tau=2,d=4}(-2\gamma_{13})\right)\right),$$

where $r_1(\ell)$ and $r_2(\ell)$ are written in appendix C.1.

The contribution of the subleading twist operators is

$$\sum_{\substack{\ell=0 \\ \text{even}}}^{\infty} \sum_i C_{\tau_{4,i}(\ell)}^2 \omega_{1,0,0}^{\tau_{4,i}(\ell),\ell,0}|_{g^3} = -\sum_{\substack{\ell=0 \\ \text{even}}} \frac{2^{\ell+1}(\ell(\ell+3) + 8)\Gamma\left(\ell + \frac{5}{2}\right)}{3\sqrt{\pi}(\gamma_{13} - 2)(\ell+1)\Gamma(\ell+3)^2}\gamma_1(\tau_4(\ell))Q_{\ell,m=0}^{\tau=4,d=4}(-2\gamma_{13}).$$

(3.37)

Expression (3.36) is cumbersome and for this reason we did not manage to compute $\gamma_1(\tau_4(\ell))$ analytically. We computed $\gamma_1(\tau_4(\ell))$ through the following numerical procedure: we can sum the series (3.36) and evaluate the orthogonality integral for $Q_{\ell,m=0}^{\tau=4,d=4}(-2\gamma_{13})$ numerically. The obtained results can be found in table 1. The integrals can be evaluated extremely efficiently and the reported errors are not due to the evaluation of the integrals. The errors are due to using the approximation (A.21) for Mack polynomials at large spins, so as to sum the series (3.36). The preceding table agrees with the expression [24]

$$\gamma_1(\tau_4(\ell)) \equiv \langle\gamma(\tau_4(\ell))\rangle|_{g^1} = \frac{24}{(\ell+1)(\ell+2) + 6}$$

(3.38)

| $\gamma_1(\tau_4(\ell))$ | Numerical Result | Analytic Expression (3.38) |
|---|---|---|
| $\ell = 0$ | $3.000000 \pm (2 \times 10^{-6})$ | $3 = 3.000000$ |
| $\ell = 2$ | $1.33334 \pm (4 \times 10^{-5})$ | $\frac{4}{3} = 1.33333$ |
| $\ell = 4$ | $0.6667 \pm (2 \times 10^{-4})$ | $\frac{2}{3} = 0.6667$ |
| $\ell = 6$ | $0.3872 \pm (6 \times 10^{-4})$ | $\frac{12}{31} = 0.3871$ |
| $\ell = 8$ | $0.250 \pm (2 \times 10^{-3})$ | $\frac{1}{4} = 0.250$ |
| $\ell = 10$ | $0.175 \pm (5 \times 10^{-3})$ | $\frac{4}{23} = 0.174$ |
| $\ell = 12$ | $0.13 \pm (2 \times 10^{-2})$ | $\frac{6}{47} = 0.13$ |

Table 1: Numerical results for the averaged anomalous dimensions of the twist-four operators $\gamma_1(\tau_4(\ell))$ at order $g^1$ and comparison with the analytic expression.

| $b_3(\ell)$ | Numerical Result |
|---|---|
| $\ell = 0$ | $-29.99998 \pm (6 \times 10^{-5})$ |
| $\ell = 2$ | $-3.7541 \pm (3 \times 10^{-4})$ |
| $\ell = 4$ | $-1.3029 \pm (8 \times 10^{-4})$ |
| $\ell = 6$ | $-0.628 \pm (1.5 \times 10^{-3})$ |
| $\ell = 8$ | $-0.358 \pm (3 \times 10^{-3})$ |
| $\ell = 10$ | $-0.226 \pm (5 \times 10^{-3})$ |
| $\ell = 12$ | $-0.152 \pm (7 \times 10^{-3})$ |
| $\ell = 14$ | $-0.107 \pm (5 \times 10^{-3})$ |

Table 2: Numerical results for the averaged corrections to the three-point functions of the twist four operators at order $g^3$.

for the averaged anomalous dimensions of twist 4 operators.

Finally, let us consider corrections to the OPE coefficients of twist 4 operators. We computed the function $b_3(\ell)$ for low spins numerically using the functional $\omega_{1,1,1}$. As before to find $b_3(\ell)$ we use the orthogonality property of Mack polynomials (A.8). By integrating $\omega_{1,1,1}|_{g^3}$ against $(8 - 2\gamma_{13})\Gamma^2(\gamma_{13})\Gamma^2(2 - \gamma_{13})Q_{\ell,m=0}^{\tau=4,d=4}(-2\gamma_{13})$ we get an equation that expresses $b_3(\ell)$ in terms of the previously found OPE data. The results that we found are presented in table 2. These predictions are new. We could not guess an analytic formula for $b_3(\ell)$. It is clear from the numerical results that $b_3(0) = -30$.

## 3.6 Order $g^4$

Let us outline the computation of the OPE coefficients of the leading twist trajectory and of $\phi^2$ at order $g^4$. We assume the formulas for $\gamma_4(j_l)$ and $a_3$.[4] We will use the $\omega_{0,0,0}$ functional. The computation is numerically more involved than at lower orders, since there are more tails to sum, as we will see next.

Let us take into account the dependence on the quantum number $m$. The sum rule can be written as $\sum_{\tau,\ell}\sum_m C_{\tau,\ell}\,\omega_{0,0,0}^{\tau,\ell,m} = 0$. For nonzero $m$, $\omega_{0,0,0}^{\tau,\ell,m}$ has double zeros for every double twist operator.

---

[4]Expression (3.21) in [25] for $\gamma_4(j_\ell)$ contains a typo. There should be $-65/96$ instead of $-65/81$. We are very grateful to Apratim Kaviraj for letting us know about it and for his precious help in navigating the $\epsilon$ expansion literature.

| Three-point function | Numerical Result | Analytic Expression |
|---|---|---|
| $c_4(\phi^2)$ | $-15.830116 \pm (2 \times 10^{-6})$ | ? |
| $c_4(2)$ | $0.0814153 \pm (3 \times 10^{-7})$ | $\frac{6037}{10368} - \frac{5\zeta(3)}{12} = 0.08141533356$ |
| $c_4(4)$ | $0.05753436 \pm (5 \times 10^{-8})$ | $\frac{1964452177}{11854080000} - \frac{9\zeta(3)}{100} = 0.05753437588$ |
| $c_4(6)$ | $0.05416485 \pm (5 \times 10^{-8})$ | $\frac{30173094509693}{298200051072000} - \frac{23\zeta(3)}{588} = 0.05416483628$ |

Table 3: Numerical and analytic results for the three-point functions of the twist two operators and $\phi^2$ at order $g^4$.

For $m = 0$, $\omega_{0,0,0}^{\tau,\ell,m}$ has double zeros at the subleading twist trajectories, $\tau = 2\Delta + 2, 2\Delta + 4, ...$ and it is nonzero for the leading twist trajectory at $\tau = 2\Delta$.

The OPE coefficients of operators of twist 6 or of higher twist are at most of order $g^4$. Since $\omega_{0,0,0}$ vanishes for the subleading twists, their contribution comes proportional to the anomalous dimensions squared, which are generically of order $g^1$. We conclude that twist 6 operators or higher do not contribute at order $g^4$. So, only twist 2 and twist 4 operators will contribute to the $\omega_{0,0,0}$ sum rule at order $g^4$.

Concerning twist 4 operators, their contribution at a given spin $l$ will come proportional to $\sum_i C_{\tau_{4,i}(\ell)}^2 \gamma^2(\tau_{4,i}(\ell))|_{g^4}$, where the index $i$ denotes the degeneracy of twist 4 operators at spin $i$. We obtained the value of this quantity from [17] (combine equations 3.6 and 3.10 in [17])

$$\sum_i C_{\tau_{4,i}(\ell)}^2 \gamma^2(\tau_{4,i}(\ell))|_{g^4} = \frac{\sqrt{\pi}2^{-2\ell-1}(\ell(\ell+3)+6)\Gamma(\ell+1)}{(\ell+1)(\ell+2)^2\Gamma(\ell+\frac{3}{2})}. \tag{3.39}$$

For this calculation, we implement a numerical scheme that is a little different from the previous cases. We consider the functional $f_4(s) = (2+s)^3(4+s)^2\omega_{0,0,0}(s)$. Afterwards, we use the orthogonality relation (A.8) for each spin $\ell$, even and odd. This will give us nontrivial equations that determine the CFT data. $f_4(s)$ has the advantage that it allows to compute the $m = 0$ contributions of the operators in the leading and subleading twist trajectories easily. Such operators contribute polynomially to the sum rule. So, we can just decompose their contribution into Mack polynomials, without needing to do the integrals (A.8) explicitly. We computed the OPE coefficients of twist 2 operators with low spins and the results that we obtained are presented in table 3. The errors come from not taking into account large spin tails. These are hard to compute because of difficulties in evaluating Mack polynomials at large spins. $c_4(\ell)$ for $\ell \geq 2$ were computed exactly in [17]. Our numerical estimates agree with the exact values which can be computed using the explicit formula presented in appendix B. The prediction for $c_4(\phi^2)$ is new.

Based on the structure of the perturbative expansion in $g$ we expect the analytic answer for $c_4(\phi^2)$ to contain $\pi^4$, $\zeta(3)$, $\zeta(5)$ with some simple rational coefficients. For example we can consider

$$1 - \frac{6\pi^4}{5} + \frac{31\zeta(3)}{24} + 95\zeta(5) = -15.83011567...$$

Our precision is not good enough to exclude or confirm various possibilities of this type.

Let us compare $c_4(\phi^2)$ that we obtained with the values from the 3d Ising model [26, 27]. In that case the relevant three-point function is $C_{\sigma\sigma\epsilon}^2 = 1.1063962(92)$. This should be matched to the

results of the $\epsilon$-expansion at $\epsilon = 1$

$$C^2_{\phi\phi\phi^2} = 2 - \frac{2\epsilon}{3} - \frac{34}{81}\epsilon^2 + \frac{1863\zeta(3) - 611}{4374}\epsilon^3 \tag{3.40}$$

$$+ \left( -\frac{6859}{472392} + \frac{323\zeta(3)}{729} - \frac{80\zeta(5)}{81} + \frac{\pi^4}{405} + \frac{2}{81}c_4(\phi^2) \right)\epsilon^4|_{\epsilon=1}$$

$$= 2.0000000 - 0.6666667 - 0.4197531 + 0.3722981 - 0.656398$$

$$= 0.62948$$

where we used that $c_4(\phi^2) = -15.830116$. Notice that the order $\epsilon^4$ correction makes the agreement with 3d Ising model worse. At order $\epsilon^2$ the deviation between $C^2_{\phi\phi\phi^2}$ and $C^2_{\sigma\sigma\epsilon}$ is $-0.1927$. At order $\epsilon^3$ it is $0.179518$. At order $\epsilon^4$ it is $-0.47688$. This is a manifestation of the asymptotic nature of the $\epsilon$-expansion. [5]

## 4 More Functionals and Holographic Applications

It is interesting to consider a broader family of functionals than the ones that we discussed so far. To do it we relax the condition that the function $F(\gamma_{12}, \gamma_{13})$ that enters (2.3) has only poles at integer locations. In particular, we consider

$$\omega_{-\gamma_{12}, p_2, p_3} = \frac{2}{(\gamma_{12} + p_2)(\gamma_{14} + p_3)}M(\gamma_{12}, \gamma_{13}) + (\text{OPE data}) = 0, \tag{4.1}$$

where we explicitly wrote the residue that involves the Mellin amplitude itself and not the OPE data. It is then convenient to define

$$\omega^{p_2, p_3}(\gamma_{12}, \gamma_{13}) \equiv \frac{(\gamma_{12} + p_2)(\gamma_{14} + p_3)}{2}\omega_{-\gamma_{12}, p_2, p_3}. \tag{4.2}$$

To get a sum rule that involves only the OPE data and not the Mellin amplitude itself we use crossing symmetry. More precisely, $\gamma_{13} \leftrightarrow \gamma_{14}$ crossing symmetry of the Mellin amplitude $M(\gamma_{12}, \gamma_{13})$ implies that

$$\omega^{p_2, p_3}(\gamma_{12}, \gamma_{13}) - \omega^{p_2, p_3}(\gamma_{12}, \gamma_{14}) = \sum_{\tau, \ell, m} C^2_{\tau, \ell}\Lambda^{p_2, p_3}_{\tau, \ell, m}(\gamma_{13}, \gamma_{14}) = 0. \tag{4.3}$$

Since $M(\gamma_{12}, \gamma_{13})$ cancels in (4.3), this sum rule involves only the OPE data.[6] This family of functionals depends on two integer parameters $p_2$ and $p_3$, and two continuous parameters $\gamma_{13}$ and $\gamma_{14}$. The function $\Lambda^{p_2, p_3}_{\tau, \ell, m}(\gamma_{13}, \gamma_{14})$ has zeros at the double twist locations $\tau = 2\Delta + 2n$. It has single zeros when when $m + n = p_2$ or $m + n = p_3$, where $m \geq 0$ and $m \in \mathbb{Z}$, and otherwise the zeros are double zeros.

We will find it useful below to consider $p_2 = p_3 = 0$. In this case the functional has a single zero at the leading double twist trajectory and double zero otherwise.[7] An example of the functional of

---

[5]Padé approximations are often used in this context (see for example [28]). We checked that a rational ansatz for $C^2_{\phi\phi\phi^2}$ with a degree 3 polynomial of $\epsilon$ in the numerator and a degree 2 denominator, which is completely fixed by the Taylor expansion (3.40) and the condition $C^2_{\phi\phi\phi^2} = \frac{1}{4}$ for $\epsilon = 2$, does not have poles for $0 < \epsilon < 2$. This approximation gives $C^2_{\phi\phi\phi^2} \approx 1.15$ for $\epsilon = 1$. This result is relatively insensitive to the $\epsilon^4$ term.

[6]The same applies to $\omega^{p_2, p_3}(\gamma_{12}, \gamma_{13}) - \omega^{\tilde{p}_2, \tilde{p}_3}(\gamma_{12}, \gamma_{14})$ but we do not consider this case in the present paper.

[7]Note that we do not have such a functional within the class $\omega_{p_1, p_2, p_3}$ discussed in the previous section.

this type was considered in [5], namely

$$\partial_x \left[ \omega^{0,0}(\frac{\Delta}{3}, \frac{\Delta}{3} - x) - \omega^{0,0}(\frac{\Delta}{3}, \frac{\Delta}{3} + x) \right] |_{x=0} = 0. \tag{4.4}$$

This functional was found to possess interesting positivity properties, namely all operators with $\tau \geq 2\Delta$ produce a nonnegative contribution.

We can similarly consider

$$\omega_{p_1,p_2,-\gamma_{14}} = \frac{2}{(\gamma_{12} + p_1)(\gamma_{12} + p_2)} M(\gamma_{12}, \gamma_{13}) + \text{OPE} = 0. \tag{4.5}$$

Again defining

$$\tilde{\omega}^{p_1,p_2}(\gamma_{12}, \gamma_{13}) \equiv \frac{(\gamma_{12} + p_1)(\gamma_{12} + p_2)}{2} \omega_{p_1,p_2,-\gamma_{14}} \tag{4.6}$$

and using crossing symmetry of the Mellin amplitude we can get sum rules in terms of the OPE data only

$$\tilde{\omega}^{p_1,p_2}(\gamma_{12}, \gamma_{13}) - \tilde{\omega}^{p_1,p_2}(\gamma_{12}, \gamma_{14}) = 0. \tag{4.7}$$

A new feature here compared to (4.3) is that by setting $p_1 = p_2 = p$ we are probing the sub-leading Polyakov condition. In this case the functional has zeros at the position of double twist operators $\tau = 2\Delta + 2n$ except at $n + m = p$, where $m \geq 0$ and $m \in \mathbb{Z}$. Next we consider a few examples of applications of these functionals.

## 4.1 $\lambda\phi^4$ in AdS at one loop

Let us consider a scalar field in AdS of dimension $\Delta$ and introduce a quartic interaction

$$\delta S_E = \frac{\lambda}{4!} \int d^d x \sqrt{g} \phi^4. \tag{4.8}$$

At tree-level this interaction leads to anomalous dimension of spin zero double twist operators [29]

$$\gamma_{n,\ell=0}^{(1)} = \lambda \frac{2^{-d-1}\pi^{-d/2}\Gamma\left(\frac{d}{2} + n\right)\Gamma(\Delta + n)\Gamma\left(\Delta - \frac{d}{2} + n + \frac{1}{2}\right)\Gamma\left(2\Delta - \frac{d}{2} + n\right)}{\Gamma\left(\frac{d}{2}\right)\Gamma(n+1)\Gamma\left(\Delta + n + \frac{1}{2}\right)\Gamma\left(\Delta - \frac{d}{2} + n + 1\right)\Gamma(2\Delta - d + n + 1)}. \tag{4.9}$$

We next consider the action of the functional (4.3) with $p_2 = p_3 = 0$ which gives [8]

$$\Lambda_{\tau,\ell,m} = 8 \frac{\gamma_{14}(\gamma_{13} + \gamma_{14} - \Delta)(\tau + 2m - \gamma_{13} - \Delta)\mathcal{Q}_{\ell,m}^{\tau,d}(-2\gamma_{13})}{(\tau + 2m - 2\gamma_{13})(\tau + 2m + 2\gamma_{14} - 2\Delta)(\tau + 2m - 2\Delta)(\tau + 2m - 2\gamma_{13} - 2\gamma_{14})}$$

$$- 8 \frac{\gamma_{13}(\gamma_{13} + \gamma_{14} - \Delta)(\tau + 2m - \gamma_{14} - \Delta)\mathcal{Q}_{\ell,m}^{\tau,d}(-2\gamma_{14})}{(\tau + 2m - 2\gamma_{14})(\tau + 2m + 2\gamma_{13} - 2\Delta)(\tau + 2m - 2\Delta)(\tau + 2m - 2\gamma_{13} - 2\gamma_{14})}. \tag{4.10}$$

Expanding to the leading order in $\lambda$, we find

$$\sum_{m,n=0}^{\infty} \frac{1}{2} C_{n,\ell=0}^2 (\gamma_{n,\ell=0}^{(1)})^2 \partial_\tau^2 \Lambda_{\tau=2\Delta+2n,\ell=0,m} + \sum_{\substack{\ell=2 \\ \text{even}}}^{\infty} C_{n=0,\ell}^2 \gamma_{0,\ell}^{(2)} \partial_\tau \Lambda_{\tau=2\Delta,\ell,m=0} = 0, \tag{4.11}$$

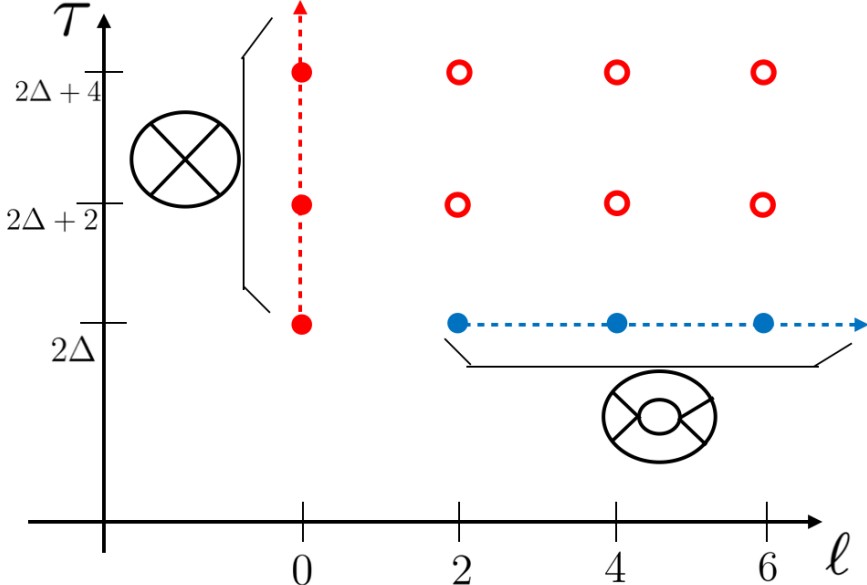

Figure 5: The functional $\Lambda_{\tau,\ell,m}$ given in (4.10), and its contribution to the sum rule (4.11). The red dots signify double zeros of the functional, $\Lambda_{\tau,\ell,m}$, and the blue dots signify *single* zeros of the functional. At subleading order, the sum rule (4.11) receives two types of contributions. A contribution at $\ell = 0$ and $n = 0, 1, 2, \ldots$ coming from contact diagrams in AdS. And a contribution from the leading tracjectory $\tau = 2\Delta$ and $\ell = 2, 4, 6, \ldots$ coming from the 1-loop bubble diagrams in AdS.

where $\gamma_{0,\ell}^{(2)}$ is the $O(\lambda^2)$ anomalous dimension of the leading double twist trajectory operators. The situation is depicted on fig. 5.

We then plug the formula (4.10) into (4.11) and note that the sum rule factorizes. In other words, it takes the form $f(\gamma_{13}) = f(\gamma_{14})$ for arbitrary $\gamma_{13}$ and $\gamma_{14}$. The solution to this of course is that $f(\gamma_{13}) = \text{const}$. Next we project this result to the collinear Mack polynomial of given spin $\ell \geq 2$ and $\tau = 2\Delta$ using (A.8). Note that the unknown constant is projected out. In this way we get the following equation

$$\gamma_{0,\ell}^{(2)} = -\frac{1 + (-1)^\ell}{2} \sum_{n=0}^{\infty} C_{n,\ell=0}^2 (\gamma_{n,\ell=0}^{(1)})^2 \mathcal{I}_{n,\ell} , \quad \ell \geq 2, \tag{4.12}$$

$$\mathcal{I}_{n,\ell} = \sum_{m=0}^{\infty} \frac{2^{4\Delta+\ell+2n-4}\Gamma(m+n+1)^2 \Gamma\left(\Delta+n+\frac{1}{2}\right)\Gamma\left(2\Delta-\frac{d}{2}+2n+1\right)}{\Gamma(m+1)\Gamma(\Delta+n)^3 \Gamma\left(2\Delta-\frac{d}{2}+m+2n+1\right)}$$

$$\times \frac{\Gamma(\Delta)^2 \Gamma\left(\Delta+\ell-\frac{1}{2}\right)}{\pi\Gamma(\Delta+\ell)\Gamma(2\Delta+\ell-1)} \int_{-i\infty}^{+i\infty} \frac{ds}{4\pi i} \frac{Q_{\ell,0}^{2\Delta,d}(s)}{\frac{s}{2}+m+n+\Delta} \Gamma^2\left(-\frac{s}{2}\right)\Gamma^2\left(\frac{s+2\Delta}{2}\right) .$$

It is possible to compute $\mathcal{I}_{n,\ell}$ precisely using the results of [30] as we explain in the next section, see equations (4.18)-(4.19). More precisely we get that

$$\mathcal{I}_{n,\ell} = -\frac{1}{C_{0,\ell}^2} \frac{(\delta hP|_{\text{pert}} + \delta hP|_{\text{nonpert}})}{(\Delta+n-\frac{\tau_\chi}{2})^2}\Big|_{\tau_\chi=2\Delta+2n} = -\frac{1}{2C_\chi^2} \frac{\gamma_\ell^{\tau_\chi,\text{exch}}}{(\Delta+n-\frac{\tau_\chi}{2})^2}\Big|_{\tau_\chi=2\Delta+2n}, \tag{4.13}$$

---

[8]From now on, we do not write the superscripts $p_2 = p_3 = 0$ and the arguments $\gamma_{13}, \gamma_{14}$ of the function $\Lambda_{\tau,\ell,m}$ to avoid cluttering.

where $\delta hP|_{\text{pert}}$ is given by (2.36) and $\delta hP|_{\text{nonpert}}$ by (2.37) in [30] upon substituting $h_i = \frac{\Delta}{2}$, $h_{\mathcal{O}} = \frac{\tau_\chi}{2}$ and $\bar{h} = \Delta + J$. By $\gamma_\ell^{\tau_\chi,\text{exch}}$ we denote the anomalous dimension induced by the tree level exchange of a scalar field $\chi$ in AdS, analyzed in the next section. The constant $C_\chi$ is the coefficient of the three point function $\langle\phi\phi\chi\rangle$ and it is proportional to the corresponding bulk cubic coupling. Eqs (4.12) and (4.13) compute the 1-loop anomalous dimension in $\lambda\phi^4$ theory, in terms of an infinite sum of tree-level anomalous dimensions in $g\phi^2\chi$ theory.[9]

We checked numerically that the formulas above agree with the conformal data from [31], who computed the 1-loop diagrams in $AdS_4$, see (5.16) there. They computed double trace anomalous dimensions for the leading twist $n = 0$ trajectory and spin $J$, for two values of external $\Delta = 1, 2$:

$$\gamma_{0,\ell}^{(2)} = -\left[\frac{4}{2\ell+1}\psi^{(1)}(\ell+1) + \frac{2}{\ell(\ell+1)}\right]\gamma^2, \quad \ell \geq 2, \quad \Delta = 1,$$

$$\gamma_{0,\ell}^{(2)} = -\frac{6}{\ell(\ell+1)(\ell+2)(\ell+3)}\gamma^2, \qquad \ell \geq 2, \quad \Delta = 2, \tag{4.14}$$

where $\gamma$ is the tree level anomalous dimension $\gamma \equiv \gamma_{n,J=0}^{(1)} = \frac{1}{8\pi^2}$, which turns out to be independent of $n$ for $\Delta = 1, 2$ and $d = 3$.

It should be possible to determine one-loop anomalous dimensions of higher twist operators $\gamma_{k,\ell}^{(2)}$ using the functional (4.3) with $p_2 + p_3 > 0$ but we do not pursue this here.

## 4.2 Tree-level scalar exchange in AdS

Consider the interaction $g\phi^2\chi$ in AdS, where $\phi$ is a scalar with scaling dimension $\Delta$ and $\chi$ is a scalar field with twist $\tau_\chi$. At tree level, the sum rule is:

$$C_\chi^2 \sum_{m=0}^\infty \Lambda_{\tau_\chi,0,m} + \sum_{\substack{\ell=2 \\ \text{even}}}^\infty C_{n=0,\ell}^2 \gamma_\ell^{\tau_\chi,\text{exch}} \partial_\tau \Lambda_{\tau=2\Delta,\ell,m=0} = 0 \tag{4.15}$$

where the functional $\Lambda_{\tau,\ell,m}$ is defined in (4.10). The first term above is the single trace exchange of $\chi$, the second term is the double trace contribution from the leading trajectory $n = 0$, and $\gamma_\ell^{\tau_\chi,\text{exch}}$ are the double-trace anomalous dimensions with $n = 0$, arising from exchange of a bulk field $\chi$. The second term above can be written explicitly as:

$$\partial_\tau \Lambda_{\tau=2\Delta,\ell,m=0} = -\frac{2^{-1-\ell}\Gamma(2\Delta+2\ell)\Gamma(2\Delta+2\ell-1)}{\Gamma(\Delta+\ell)^4\Gamma(2\Delta+\ell-1)}\left(Q_{\ell,0}^{2\Delta,d}(-2\gamma_{13}) - Q_{\ell,0}^{2\Delta,d}(-2\gamma_{14})\right) \tag{4.16}$$

The equation above depends on $\gamma_{13}$ only through $Q_{\ell,0}^{2\Delta,d}(-2\gamma_{13})$. Thus, we can extract $\gamma_\ell^{\tau_\chi,\text{exch}}$ by integrating (4.15) against $\int \frac{ds}{4\pi i}\Gamma^2(-\frac{s}{2})\Gamma^2(\frac{s+\tau}{2})Q_{\ell,0}^{2\Delta,d}(s)$ with $s = -2\gamma_{13}$, and using orthogonality of the Mack polynomials. This gives:

$$\gamma_\ell^{\tau_\chi,\text{exch}} = \frac{-C_\chi^2\,\Gamma^2(\Delta)\Gamma(\Delta+\ell-\frac{1}{2})}{\sqrt{\pi}2^{2-\ell-2\Delta}\Gamma(\Delta+\ell)\Gamma(2\Delta+\ell-1)}\int_{-i\infty}^{i\infty}\frac{ds}{4\pi i}\Gamma^2\left(-\frac{s}{2}\right)\Gamma^2\left(\frac{s+2\Delta}{2}\right)Q_{\ell,0}^{2\Delta,d}(s)\sum_{m=0}^\infty \Lambda_{\tau_\chi,0,m} \tag{4.17}$$

---

[9]This is reminiscent of the identity $[G_\Delta(X,Y)]^2 = \sum_{n=0}^\infty a_n(\Delta)G_{2\Delta+2n}(X,Y)$ that expresses the square of the AdS bulk-to-bulk scalar propagator $G_\Delta(X,Y)$ as a sum of propagators with double-trace dimensions [29]. This identity can be used to write the 1-loop diagrams of $\lambda\phi^4$ theory in AdS in terms of tree level exchange diagrams, which is what formulas (4.12) and (4.13) achieve.

where $\ell \geq 2$. This equation computes the $n = 0$ double trace anomalous dimension arising from a tree level exchange of a scalar $\chi$. Plugging Eq. (4.10) inside (4.17) gives:

$$\gamma_\ell^{\tau_\chi,\text{exch}} = \frac{C_\chi^2 2^{\ell+2\Delta-2}\Gamma^2(\Delta)\Gamma(\Delta+\ell-\frac{1}{2})}{\pi^{\frac{5}{2}}\Gamma(\Delta+\ell)\Gamma(2\Delta+\ell-1)} \frac{\Gamma(\tau_\chi)\sin^2(\pi(\Delta-\frac{\tau_\chi}{2}))}{\Gamma^4(\frac{\tau_\chi}{2})} \sum_{m=0}^\infty \frac{\Gamma^2(\frac{2m-2\Delta+\tau_\chi+2}{2})}{\Gamma(m+1)(\tau_\chi-\frac{d}{2}+1)_m}$$

$$\times(-1)\int_{-i\infty}^{i\infty} \frac{ds}{4\pi i} \mathcal{Q}_{\ell,0}^{2\Delta,d}\, \Gamma^2\left(-\frac{s}{2}\right)\Gamma^2\left(\frac{s+2\Delta}{2}\right)\left(\frac{1}{s+2m+\tau_\chi} + \frac{1}{-s+2m-2\Delta+\tau_\chi}\right)(4.18)$$

In principle, this computation can be generalized to exchanges of operators with spin $J > 0$. The main difference is that the associated tree-level Mellin amplitude grows like $\gamma_{12}^{J-1}$ in the Regge limit. This means that we need to take into account the contribution from the arcs at infinity in (2.3) or choose a function $F$ decaying faster than $1/\gamma_{12}^{J+1}$ at infinity.

Note that the RHS of (4.17) has double zeros at $\tau_\chi = 2\Delta + 2n$, due to the factor $\sin^2(\pi(\Delta-\frac{\tau_\chi}{2}))$. We now take the limit $\tau_\chi \to 2\Delta + 2n$ of Eq. (4.18), and notice that we get:

$$-\frac{1}{2C_\chi^2}\frac{\gamma_\ell^{\tau_\chi,\text{exch}}}{(\Delta+n-\frac{\tau_\chi}{2})^2}\Big|_{\tau_\chi\to2\Delta+2n} = \mathcal{I}_{n,\ell} \tag{4.19}$$

where $\mathcal{I}_{n,\ell}$ is given in (4.12). In other words, the coefficients of the double zeros are proportional to $\mathcal{I}_{n,\ell}$. We used this result in equation (4.13).

We could also study $\phi^3$ theory at 1-loop level. Notice that this involves Witten diagrams, like the box diagram in AdS, that are not present in $\phi^4$ theory. Using methods similar to the ones in section 4.1, one should be able to derive formulas for the one-loop anomalous dimensions of the leading twist operators $\gamma_{0,\ell}^{(2)}$. We leave a detailed analysis of this model for the future.

## 4.3 $(\partial\phi)^4$ in AdS

Let us next consider a weakly coupled theory in AdS with the following low energy action

$$S_E = \int d^{d+1}x\sqrt{g}\left[\frac{1}{2}(\partial_\mu\phi)^2 + \lambda(\partial_\mu\phi\partial^\mu\phi)^2 + ...\right], \tag{4.20}$$

where $\lambda$ is a small parameter and ... stands for terms that are higher order in $\lambda$.

We can study this theory with the functional (4.4) which was also considered in [5]. It takes the following form

$$\partial_x\left[\omega^{0,0}(\frac{\Delta}{3},\frac{\Delta}{3}-x) - \omega^{0,0}(\frac{\Delta}{3},\frac{\Delta}{3}+x)\right]|_{x=0} = \sum_{\tau,\ell,m} C_{\tau,\ell}^2\alpha_{\tau,\ell,m} = 0, \tag{4.21}$$

$$\alpha_{\tau,\ell,m} = -\frac{16\Delta}{3(\tau-\frac{2\Delta}{3}+2m)(\tau-\frac{4\Delta}{3}+2m)}\left(\frac{(\tau+2m-\Delta)\mathcal{Q}_{\ell,m}^{\tau,d}(\gamma_{13}=\frac{\Delta}{3})}{(\tau-\frac{2\Delta}{3}+2m)(\tau-\frac{4\Delta}{3}+2m)} - \frac{\Delta}{3}\frac{\partial_{\gamma_{13}}\mathcal{Q}_{\ell,m}^{\tau,d}(\gamma_{13}=\frac{\Delta}{3})}{\tau+2m-2\Delta}\right).$$

At tree level, the contact diagram contributes only to the OPE data of the double twist operators with spin $\ell = 0$ and $\ell = 2$. Since the functional (4.21) has double zeros at the position of double twist operators for $\ell = 0$, the only contribution at order $\lambda$ comes from $\ell = 2$ and $\tau = 2\Delta$ (i.e $n = 0$).

Operators with $n \geq 1$ contribute at order $O(\lambda^2)$. The sum rule at order $O(\lambda)$ therefore takes the following form

$$\frac{2\Delta(\Delta+1)(2\Delta+1)\Gamma(2\Delta+4)}{3\Gamma(\Delta+2)^4} C^2_{n=0,\ell=2} \gamma^{(1)}_{n=0,\ell=2} + \sum_{\tau \geq 2\Delta,\ell,m} C^2_{\tau,\ell} \alpha_{\tau,\ell,m}\big|_{\lambda^1} = 0. \tag{4.22}$$

Non-negativity of the functional $\alpha_{\tau \geq 2\Delta,\ell,m}$ then implies non-positivity of the anomalous dimension

$$\boxed{\gamma^{(1)}_{n=0,\ell=2} \leq 0} \tag{4.23}$$

This matches the previously derived causality constraint from [32, 33]. As discussed in [5] the argument above only applies to $\frac{d-2}{2} < \Delta < \frac{3(d-2)}{4}$, where our functional converges. We do not improve this range in the present paper.

Let us now briefly discuss a possible mechanism how (4.22) can be satisfied. In order for it to work, there must be a cancelation between the tree level result $\sim \gamma^{(1)}_{n=0,\ell=2}$ and heavy operators $\tau \geq 2\Delta$. We want to estimate the contribution of the heavy operators into the sum rule. At large energies the theory becomes non-perturbative, we can estimate this scale by looking at the anomalous dimensions $\gamma^{(1)}_{n,\ell=2}$ at large $n$ and demanding that $\gamma^{(1)}_{n,\ell} \sim 1$. Using dimensional analysis we can immediately write[10], we have

$$\gamma^{(1)}_{n,\ell=0,2} \sim \lambda(\text{energy})^{d+1} \sim \lambda n^{d+1}. \tag{4.24}$$

Therefore, the value of $n$ above which non-perturbative effects become important is:

$$n_* \sim \lambda^{\frac{-1}{d+1}} \qquad \rightarrow \qquad \tau_* \sim 2n_* \sim \lambda^{\frac{-1}{d+1}} \tag{4.25}$$

Using the result from appendix F of [5], we can estimate the expected contribution to the sum rule at large $\tau$ and fixed $\ell$ as

$$\sum_{\tau \geq 2\Delta,m} C^2_{\tau,\ell}\alpha_{\tau,\ell,m}|_{\lambda^1} \sim \int_{\tau_*}^{\infty} \frac{d\tau}{\tau^{d+2}} \sim \int_{\lambda^{\frac{-1}{d+1}}}^{\infty} \frac{d\tau}{\tau^{d+2}} \sim \frac{1}{\tau^{d+1}}\Big|_{\lambda^{\frac{-1}{d+1}}}^{\infty} \sim \lambda. \tag{4.26}$$

This estimate is therefore consistent with the structure of the sum rule (4.22). In the estimate above we assumed that the universal OPE asymptotic derived in [34] for fixed spin OPE data does not change in the presence of $\sin^2\frac{\pi(\tau-2\Delta)}{2}$. In the terminology of [18] we assumed that the UV completion of (4.20) is opaque. It would be interesting to understand what are other possibilities to satisfy (4.22).

## 4.4 UV Complete Holographic Theories

More generally, we can consider a theory with a weakly coupled gravity dual. As in the previous section we can compute the tree-level Mellin amplitude and we will find that

$$\lim_{\gamma_{12}\to\infty} M_{\text{tree}}(\gamma_{12},\gamma_{13}) = \gamma_{12}^2 f_{\text{IR}}(\gamma_{13})(1 + O(\gamma_{12}^{-1})) \sim O\left(\frac{1}{c_T}\right), \tag{4.27}$$

where $c_T$ is the central charge of the CFT and it is inversely proportional to the AdS gravitational coupling. In the formula above $f_{\text{IR}}(\gamma_{13})$ receives contributions from the exchanges by spin two

---

[10]The coupling constant has mass dimension $[\lambda] = -d - 1$, and $\gamma_{n,\ell}$ is dimensionless.

particles, e.g. graviton exchange, as well as tree-level higher derivative interactions $(\partial_\mu \phi \partial^\mu \phi)^2$ considered in the previous section, as well as from $\phi^2 \phi_{;\mu\nu\sigma} \phi^{;\mu\nu\sigma}$, see [23], which together contribute as $f_{\text{hd}}(\gamma_{13}) = c_1 + c_2 \gamma_{13}$. On the other hand, spin zero and spin one particle exchanges will not contribute to (4.27).

Let us also present for completeness the result for the graviton exchange, see formula (164) in [35],

$$f_{\text{grav}}(\gamma_{13}) = C^2_{T_{\mu\nu}} \frac{(d-1)d\Gamma(d+2)\,_3F_2\left(\frac{d}{2}-\Delta, \frac{d}{2}-\Delta, \frac{d}{2}+\gamma_{13}-\Delta-1; \frac{d}{2}+1, \frac{d}{2}+\gamma_{13}-\Delta; 1\right)}{32\Gamma\left(\frac{d}{2}+1\right)^4 \Gamma\left(-\frac{d}{2}+\Delta+1\right)^2 (d-2\Delta+2\gamma_{13}-2)}.$$
(4.28)

If we now consider a function

$$F(\gamma_{12}, \gamma_{13}) = \frac{1}{\gamma_{12}^3}\left(1 + O(\gamma_{12}^{-1})\right)$$
(4.29)

in the functional (2.3) we will find

$$\oint_{\mathcal{C}_\infty} \frac{d\gamma_{12}}{2\pi i} M_{\text{tree}}(\gamma_{12}, \gamma_{13}) F(\gamma_{12}, \gamma_{13}) = f_{\text{IR}}(\gamma_{13}).$$
(4.30)

On the other hand, by closing the contour on the singularities of $M_{\text{tree}}$ and $F(\gamma_{12}, \gamma_{13})$ we will get the type of sum rules that we analyzed in the paper, namely

$$C^2_{T_{\mu\nu}} \sum_{m=0}^{\infty} \omega_F^{d,2,m} + \sum_{n=0}^{n_{\max}} \sum_{\ell,m} C^2_{\tau(n,\ell),\ell} \, \omega_F^{\tau(n,\ell),\ell,m} = f_{\text{IR}}(\gamma_{13}),$$
(4.31)

where the second term in the LHS of (4.31) represents the contribution of the double trace operators. For simple choices of meromorphic $F$ that we consider here only a finite number of double trace families contribute at leading order in $\frac{1}{c_T}$. This is signified by $n_{\max}$ in the sum above.[11]

Consider now a UV complete theory which at low energies is given by $M_{\text{tree}}(\gamma_{12}, \gamma_{13})$. We can apply to this theory a functional $\omega_F$ to get a sum rule

$$\omega_F = \sum_{\tau,\ell,m} C^2_{\tau,\ell} \omega_F^{\tau,\ell,m} = 0.$$
(4.32)

Imagine we now want to analyze this sum rule to leading order in $\frac{1}{c_T}$. The contribution of $M_{tree}$ to the sum rule (4.32) can be conveniently computed using (4.31). Of course, we can alternatively sum over the relevant single and double trace operator OPE data that contributes at order in $\frac{1}{c_T}$ in (4.32) however it is much simpler to use (4.31) instead. In this way we get that (4.32) becomes

$$f_{\text{IR}}(\gamma_{13}) + \sideset{}{'}\sum_{\tau,\ell,m} C^2_{\tau,J} \omega_F^{\tau,J,m}|_{\frac{1}{c_T}} = 0,$$
(4.33)

where by $\sum'$ we denoted the contribution of all operators that are perturbatively suppressed and are responsible for the fact that the sum rule is satisfied in the non-perturbative theory. We can

---

[11]For example, given $F(\gamma_{12}, \gamma_{13})$ that satisfies (4.29), one can consider a theory $\phi^2\chi$ and explicitly check that the correction to the OPE data of double trace operators exactly cancels the contribution of the operator dual to $\chi$ in agreement with the general argument above.

think of these operators as the UV completion of the theory. For example, in the theory of a scalar minimally coupled to gravity $f_{\mathrm{IR}}(\gamma_{13}) = f_{\mathrm{grav}}(\gamma_{13})$. The argument above can be repeated for more general $F(\gamma_{12}, \gamma_{13}) \sim \frac{1}{\gamma_{12}^{2k+1}}$. This would lead to similar sum rules which are sensitive to the $\frac{1}{c_T^k}$ terms in the large $c_T$ expansion.

An interesting situation is when each term in $\sum'$ is non-negative. We discussed some example of such functionals in the present paper, see e.g. section 4.3. The sum rule (4.33) then is an interesting prediction about the UV completion of a given theory. It would be very interesting to understand a detailed mechanism how such sum rules are satisfied in the UV completion of gravitational theories and if it imposes any new nontrivial constraint on the consistent low energy effective theories. We leave this interesting question for the future work.

## Acknowledgements

We are very grateful for discussions with Apratim Kaviraj and Johan Henriksson. DC, JP and JS are supported by the Simons Foundation grant 488649 (Simons Collaboration on the Nonperturbative Bootstrap) and by the Swiss National Science Foundation through the project 200021-169132 and through the National Centre of Competence in Research SwissMAP.

## A    Notes on Mack Polynomials

We denote the residue of the Mellin amplitude $M(\gamma_{12}, \gamma_{13})$ as

$$M(\gamma_{12}, \gamma_{13}) \approx -\frac{1}{2} \frac{C_{\tau,\ell}^2 \mathcal{Q}_{\ell,m}^{\tau,d}(-2\gamma_{13})}{\gamma_{12} - (\Delta - \frac{\tau}{2} - m)}. \tag{A.1}$$

The calligraphic $\mathcal{Q}_{\ell,m}^{\tau,d}(s)$ is related to the usual Mack polynomial $Q_{\ell,m}^{\tau,d}(s)$ as follows

$$\mathcal{Q}_{\ell,m}^{\tau,d}(s) = -K(\tau,\ell,m) Q_{\ell,m}^{\tau,d}(s), \tag{A.2}$$

where the proportionality factor is given by

$$K(\tau,\ell,m) \equiv \frac{2(\ell+\tau-1)_\ell \Gamma(2\ell+\tau)}{2^\ell \Gamma\left(\frac{1}{2}(2\ell+\tau)\right)^4 \Gamma(m+1) \Gamma\left(-m+\Delta-\frac{\tau}{2}\right)^2 \left(-\frac{d}{2}+\ell+\tau+1\right)_m}. \tag{A.3}$$

Note the minus sign in (A.2).

For the Mack polynomial $Q_{\ell,m}^{\tau,d}(s)$ we use the following representation [36]

$$Q_{\ell,m}^{\tau,d}(s) = 4^\ell (-1)^\ell \sum_{n_1=0}^{\ell} \sum_{m_1=0}^{\ell-n_1} (-m)_{m_1} \left(m + \frac{s}{2} + \frac{\tau}{2}\right)_{n_1} \mu(\ell, m_1, n_1, \tau, d), \tag{A.4}$$

where

$$\mu(\ell, m, n, \tau, d) \equiv \frac{2^{-\ell} \Gamma(\ell+1)(-1)^{m+n} \left(\frac{d}{2}+\ell-1\right)_{-m} \left(\ell-m+\frac{\tau}{2}\right)_m \left(n+\frac{\tau}{2}\right)_{\ell-n} \left(m+n+\frac{\tau}{2}\right)_{\ell-m-n}}{\Gamma(m+1)\Gamma(n+1)\Gamma(\ell-m-n+1)}$$

$$(2\ell+\tau-1)_{n-\ell} \times_4 F_3 \left(-m, -\frac{d}{2}+\frac{\tau}{2}+1, -\frac{d}{2}+\frac{\tau}{2}+1, \ell+n+\tau-1; \ell-m+\frac{\tau}{2}, n+\frac{\tau}{2}, -d+\tau+2; 1\right). \tag{A.5}$$

## A.1 Mack Polynomial Projections

We will also find it very useful to consider functionals that are obtained by integrating $\omega_{p_1,p_2,p_3}(\gamma_{13})$ against Mack polynomials. To agree with the standard conventions for Mack polynomials we switch to the $s$ variable

$$s \equiv -2\gamma_{13}. \tag{A.6}$$

We then consider the following projection

$$\omega_{p_1,p_2,p_3}^{(\tau,\ell)} \equiv \int_{-i\infty}^{+i\infty} \frac{ds}{2\pi i} \omega_{p_1,p_2,p_3}(s) Q_{\ell,0}^{\tau,d}(s) \Gamma^2\left(-\frac{s}{2}\right) \Gamma^2\left(\frac{s+\tau}{2}\right). \tag{A.7}$$

where the reduced Mack polynomials $Q_{\ell,m}^{\tau,d}(s)$ were defined above. Note that $Q_{\ell,0}^{\tau,d}(s)$ is $d$-independent. The collinear Mack polynomials $Q_{\ell,0}^{\tau,d}(s)$ satisfy the orthogonality relation

$$\int_{-i\infty}^{+i\infty} \frac{ds}{4\pi i} Q_{\ell,0}^{\tau,d}(s) Q_{\ell',0}^{\tau,d}(s) \Gamma^2\left(-\frac{s}{2}\right) \Gamma^2\left(\frac{s+\tau}{2}\right) = \frac{\delta_{\ell,\ell'}(-1)^\ell 4^\ell \Gamma(\ell+1)\Gamma\left(\ell+\frac{\tau}{2}\right)^4}{(2\ell+\tau-1)(\ell+\tau-1)_\ell^2 \Gamma(\ell+\tau-1)}. \tag{A.8}$$

Another useful identity is the following

$$\int_{-i\infty}^{+i\infty} \frac{ds}{4\pi i} Q_{\ell,0}^{\tau,d}(s) \partial_\tau Q_{\ell',0}^{\tau,d}(s) \Gamma^2\left(-\frac{s}{2}\right) \Gamma^2\left(\frac{s+\tau}{2}\right)$$

$$= \begin{cases} 0 & \ell' \leq \ell \,, \\ \frac{(-1)^\ell 2^{4-\ell-\ell'-2\tau}\pi}{(\ell'-\ell)(\ell'+\ell+\tau-1)} \frac{\Gamma(\ell+\frac{\tau}{2})\Gamma(\ell'+\frac{\tau}{2})\Gamma(\ell+\tau-1)\Gamma(\ell'+1)}{\Gamma(\ell+\frac{\tau-1}{2})\Gamma(\ell'+\frac{\tau-1}{2})} & \ell' > \ell \,. \end{cases} \tag{A.9}$$

## A.2 Proof of Positivity of $m = 0$ Mack polynomials

The $m = 0$ Mack polynomials are given by the formula

$$Q_{\ell,0}^{\tau,d} = \frac{2^\ell((\tau/2)_\ell)^2}{(\tau+\ell-1)_\ell} {}_3F_2(-\ell, \ell+\tau-1, -\frac{s}{2}; \frac{\tau}{2}, \frac{\tau}{2}; 1) \tag{A.10}$$

Mack polynomials are related to continuous Hahn polynomials, which obey useful recursion relations to study their positivity properties. A continuous Hahn polynomial is defined by[12]

$$p_n(x; a, b, c, d) = i^n \frac{(a+c)_n (b+d)_n}{\Gamma(n+1)} {}_3F_2(-n, n+a+b+c+d-1, a+ix; a+c, a+d; 1) \tag{A.11}$$

Continuous Hahn polynomials obey a recursion relation. Let us define

$$p_n(x) = p_n(x; a, b, c, d) \frac{\Gamma(n+1)}{(n+a+b+c+d-1)_n}, \tag{A.12}$$

$$A_n = -\frac{(n+a+b+c+d-1)(n+a+c)(n+a+d)}{(2n+a+b+c+d-1)(2n+a+b+c+d)}, \tag{A.13}$$

$$C_n = \frac{n(n+b+c-1)(n+b+d-1)}{(2n+a+b+c+d-2)(2n+a+b+c+d-1)}. \tag{A.14}$$

---

[12]We took formulas for continuous Hahn polynomials from the book [37], see the pages 200 and 201.

Then,

$$xp_n(x) = p_{n+1}(x) + i(A_n + C_n + a)p_n(x) - A_{n-1}C_n p_{n-1}(x). \tag{A.15}$$

We have that

$$p_\ell(0) = 2^{-\ell} i^\ell Q_{\ell,0}^{\tau,d}, \tag{A.16}$$

provided we pick

$$a = -\frac{s}{2}, \quad b = -\frac{s}{2}, \quad c = \frac{s+\tau}{2}, \quad d = \frac{s+\tau}{2}. \tag{A.17}$$

Let us define $\tilde{Q}(\ell) = 2^{-\ell} Q_{\ell,0}^{\tau,d}(s)$. Then,

$$-\frac{\ell(-2+\ell+\tau)(-2+2\ell+\tau)^2}{16(-3+2\ell+\tau)(-1+2\ell+\tau)}\tilde{Q}(\ell-1) - \frac{2s+\tau}{4}\tilde{Q}(\ell) + \tilde{Q}(\ell+1) = 0. \tag{A.18}$$

Let us use this recursion relation to demonstrate that $m = 0$ Mack polynomials are positive, i.e. $\tilde{Q}(\ell) \geq 0$, provided $s \geq -\frac{\tau}{2}$. For spins $\ell = 0, 1, 2$:

$$\tilde{Q}(0) = 1, \quad \tilde{Q}(1) = \frac{2s+\tau}{4}, \quad \tilde{Q}(2) = \frac{1}{16}\left(4s^2 + 4s\tau + \frac{\tau^2(\tau+2)}{\tau+1}\right) \tag{A.19}$$

By mathematical induction the recursion relation implies the positivity at all positive integer $\ell$. Furthermore the recursion relation

$$-\frac{1}{2}\tilde{Q}(\ell) - \frac{\ell(-2+\ell+\tau)(-2+2\ell+\tau)^2}{16(-3+2\ell+\tau)(-1+2\ell+\tau)}\partial_s\tilde{Q}(\ell-1) - \frac{2s+\tau}{4}\partial_s\tilde{Q}(\ell) + \partial_s\tilde{Q}(\ell+1) = 0$$

together with

$$\partial_s\tilde{Q}(0) = 0, \quad \partial_s\tilde{Q}(1) = \frac{1}{2} \quad \partial_s\tilde{Q}(2) = \frac{1}{4}(2s+\tau) \tag{A.20}$$

implies that $\partial_s\tilde{Q}(\ell)$ is positive for $s \geq -\frac{\tau}{2}$. We believe that this argument can be generalized to establish positivity of higher derivatives of Mack polynomials, but we have not tried to do that.

## A.3 Limits of Mack polynomials

Let us study Mack polynomials when the spin $\ell$ is much bigger than all the other quantum numbers. The following expansion

$$Q_{\ell,m}^{\tau,d}(s) \approx \frac{\sqrt{\pi}\ell^{s+\frac{1}{2}}2^{-\ell-m-\tau+2}\Gamma\left(\ell+\frac{\tau}{2}\right)^2\Gamma(\ell+\tau-1)p_m(s)\Gamma\left(-\frac{d}{2}+\ell+m+\tau+1\right)}{\Gamma\left(\ell+\frac{\tau}{2}-\frac{1}{2}\right)\Gamma\left(\ell+\frac{\tau}{2}+\frac{1}{2}\right)\Gamma\left(-\frac{d}{2}+\ell+\tau+1\right)\Gamma\left(\frac{1}{2}(2m+s+\tau)\right)^2} \tag{A.21}$$

$$+\frac{\sqrt{\pi}(-1)^\ell 2^{-\ell-m-\tau+2}\Gamma\left(\ell+\frac{\tau}{2}\right)^2\Gamma(\ell+\tau-1)\ell^{-2m-s-\tau+\frac{1}{2}}p_m(-2m-s-\tau)\Gamma\left(-\frac{d}{2}+\ell+m+\tau+1\right)}{\Gamma\left(-\frac{s}{2}\right)^2\Gamma\left(\ell+\frac{\tau}{2}-\frac{1}{2}\right)\Gamma\left(\ell+\frac{\tau}{2}+\frac{1}{2}\right)\Gamma\left(-\frac{d}{2}+\ell+\tau+1\right)}$$

$$+O\left(\frac{1}{\ell}\right), \quad \ell \gg 1 .$$

can be derived [38] from the recursion relation in $m$, see (A.24). $p_m(s)$ is an $m$-th degree polynomial in $s$ and it obeys $p_m(s) = s^m + \mathcal{O}(\frac{1}{s})$. We found that

$$p_0(s) = 1 , \quad p_1(s) = \frac{d-2}{2} + s + \frac{\tau}{2}. \tag{A.22}$$

### A.4 Some facts about Mack polynomials

Mack polynomials obey the following symmetry property

$$Q_{\ell,m}^{\tau,d}(s) = (-1)^\ell Q_{\ell,m}^{\tau,d}(-s-\tau-2m) \ . \tag{A.23}$$

For this reason, at $s = -\frac{\tau}{2} - m$ odd spin Mack polynomials vanish and even spin Mack polynomials have a vanishing derivative. Mack polynomials obey the following recursion relation in $m$ for fixed $\tau$ and $\ell$ [39]

$$Q(s,m)\left(-4dm - 4\ell^2 - 4\ell(\tau-1) + 4m^2 - 4ms + 4m\tau - 2s^2 - 2s\tau - \tau^2\right) \tag{A.24}$$
$$+ 2mQ(s,m-1)(d-2(\ell+m+\tau)) + 2mQ(s+2,m-1)(d-2(\ell+m+\tau))$$
$$+ s^2Q(s-2,m) + (2m+s+\tau)^2Q(s+2,m) = 0.$$

We also found it useful to express $m > 0$ Mack polynomials in terms of $m = 0$ Mack polynomials. In $d = 4$ the relevant formulas are

$$Q_{\ell,m=1}^{\tau,d=4}(s) = \left(s + \frac{\tau}{2} + 1\right)Q_{\ell-1,m=0}^{2+\tau,d=4}(s) \tag{A.25}$$

and

$$Q_{\ell,m=2}^{\tau,d=4}(s) = a_1 Q_{\ell-4,m=0}^{4+\tau,d=4}(s) + a_2 Q_{\ell-3,m=0}^{4+\tau,d=4}(s) \tag{A.26}$$

where

$$a_1 = \frac{(\ell-3)(\ell+\tau-1)(2\ell+\tau-4)^2\left(4s^2(\tau-1) + 4s\left(\tau^2+3\tau-4\right) + \tau^3 + 6\tau^2 + 8\tau - 16\right)}{16(\tau-1)(2\ell+\tau-5)(2\ell+\tau-3)}, \tag{A.27}$$

$$a_2 = \frac{8s^3(\tau-1) + 12s^2\left(\tau^2+3\tau-4\right) + s\left(6\tau^3+40\tau^2+48\tau-96\right) + \tau^4 + 10\tau^3 + 32\tau^2 + 16\tau - 64}{8(\tau-1)}. \tag{A.28}$$

We believe similar recursion relations can be generated at any finite $m$ and in any $d$.

## B OPE data

For our purposes, it is important to know what has been computed before. The dimensions of $\phi$ and $\phi^2$ are known to the order $\epsilon^5$ [40]. The dimensions of the leading twist trajectory (which starts at spin 2) are known to order $\epsilon^4$ [25]. As to the subleading twist trajectory, twist 4 operators are degenerate for a given spin. This degeneracy was analysed in [22,41] to order $\epsilon^2$, where anomalous dimensions and OPE coefficients are given for operators with low spins. However, no formulas for arbitrary spin are given. We conjecture a formula for the averaged anomalous dimensions of twist 4 operators (3.38).

The OPE coefficient of $\phi^2$ with two operators $\phi$ is known to order $\epsilon^3$ [14]. We compute the order $\epsilon^4$ correction (see table 3). The OPE coefficients of the leading twist trajectory are known to the order $\epsilon^4$ [17]. A formula for the averaged OPE coefficients of twist 4 operators is known to the order $\epsilon$ [17].

Using our notation from section 3.1, let us register the value of known quantities in the $\epsilon$ expansion. The spacetime dimensionality $d$ defined in (3.2) takes the form

$$d = 4 - 3g + \frac{8}{3}g^2 + \left(-12\zeta(3) - \frac{23}{12}\right)g^3 + \left(\frac{75\zeta(3)}{2} + 120\zeta(5) - \frac{77}{48} - \frac{3\pi^4}{10}\right)g^4 + a_5 g^5 + \dots \,,$$

(B.1)

where for completeness we also note here $a_5$ that was computed in [40] even though we did not use it in our paper

$$a_5 = -\frac{159\zeta(3)}{2} - 72\zeta(3)^2 - 504\zeta(5) - 1323\zeta(7) + \frac{4175}{576} + \frac{289\pi^4}{240} + \frac{10\pi^6}{21} \,.$$

(B.2)

Similarly for the dimensionality of the scalar defined in (3.3) takes the following form

$$\Delta_\phi - \frac{d-2}{2} = \frac{1}{12}g^2 + \frac{5}{48}g^3 - \frac{7}{192}g^4 + \left(\frac{7\zeta(3)}{16} + \frac{1}{2304}\right)g^5 + \dots \,.$$

(B.3)

For the twist-two operators the OPE data was defined in (3.4). The anomalous dimensions that are known but were not explicitly written in the main text take the following form

$$\gamma_2(j_\ell) = -\frac{1}{\ell(\ell+1)}, \quad \gamma_3(j_\ell) = -\frac{2S_1(\ell)}{\ell(\ell+1)} + \frac{3\left(2\ell^2 - 1\right)}{2\ell^2(\ell+1)^2} \,,$$

$$\gamma_4(j_\ell) = -\frac{3S_2(\ell) + 2S_1^2(\ell)}{\ell(\ell+1)} + \frac{\ell\left(\ell^3 + 2\ell^2 - 3\ell - 4\right)}{4\ell^3(\ell+1)^3}\left(S_2(\frac{\ell-1}{2}) - S_2(\frac{\ell}{2}) + \frac{\pi^2}{3}\right) + \frac{(53\ell^2 + 17\ell - 18)S_1(\ell)}{6\ell^2(\ell+1)^2}$$

$$+ \frac{1}{12\ell^3(\ell+1)^3}\left(-58\ell^4 + 26\ell^3 + 81\ell^2 - 15\ell - 33\right) \,.$$

(B.4)

For the three-point functions of the twist-two operators that were defined in (3.8) we have

$$c_2(j_\ell) = \frac{S_1(2\ell) - S_1(\ell) + \frac{1}{\ell+1}}{\ell(\ell+1)},$$

$$c_3(j_\ell) = \frac{3(\ell+\frac{1}{2})(S_1(2\ell)-\frac{\ell-1}{\ell+1})-(\ell+\frac{3}{2})S_1(\ell)}{\ell^2(\ell+1)^2} + \frac{3(S_1(\ell)-S_1(2\ell)+S_2(2\ell))-2\left(S_1^2(\ell)-S_1(\ell)S_1(2\ell)+S_2(\ell)\right)}{\ell(\ell+1)},$$

$$c_4(j_\ell) = \frac{2\ell\left(29\ell^3 - 9(\ell+1)^2\left(\ell^2+\ell+4\right)\zeta(3) - 48\ell^2 - 38\ell + 24\right) + 33}{12\ell^3(\ell+1)^4}$$

$$- \frac{2S_1^3(\ell)}{\ell^2+\ell} + \frac{S_1^2(2\ell)}{2\ell^2(\ell+1)^2} + S_1^2(\ell)\big(\frac{2S_1(2\ell)}{\ell^2+\ell} + \frac{\ell(53\ell+29)-15}{6\ell^2(\ell+1)^2}\big)$$

$$+ \frac{(\ell(\ell(\ell+2)-7)-4)S_2(\frac{\ell}{2}))}{2\ell^3(\ell+1)^3} + \frac{(\ell(\ell(\ell(89\ell+130)-10)+45)+48)S_2(\ell)}{12\ell^3(\ell+1)^3}$$

$$+ S_1(2\ell)\left(\frac{\ell\left(\ell\left(58\ell^2-26\ell-81\right)+27\right)+33}{12\ell^3(\ell+1)^3} + \frac{\left(\ell^2+\ell-4\right)S_2(\frac{\ell}{2})}{2\ell^2(\ell+1)^2} + \frac{2\left(\ell^2+\ell+2\right)S_2(\ell)}{\ell^2(\ell+1)^2}\right)$$

$$+ \frac{(30-\ell(71\ell+17))S_2(2\ell)}{6\ell^2(\ell+1)^2} + \frac{\left(\ell^2+\ell-4\right)S_3(\frac{\ell}{2})}{4\ell^2(\ell+1)^2} + \frac{(4-7\ell(\ell+1))S_3(\ell)}{\ell^2(\ell+1)^2} + \frac{9S_3(2\ell)}{\ell^2+\ell}$$

$$+ S_1(\ell)\big(\frac{\ell(\ell(83-2\ell(29\ell+40))+9)-33}{12\ell^3(\ell+1)^3} + \frac{(12-\ell(53\ell+17))S_1(2\ell)}{6\ell^2(\ell+1)^2} - \frac{\left(\ell^2+\ell-4\right)S_2(\frac{\ell}{2})}{2\ell^2(\ell+1)^2}$$

$$+ \frac{(-6\ell(\ell+1)-4)S_2(\ell)}{\ell^2(\ell+1)^2} + \frac{6S_2(2\ell)}{\ell^2+\ell}\big).$$

# C   Auxiliary formulas

## C.1   Miscellaneous formulas

Here we present a few bulky formulas to which we refer from the main part of the paper.

$$r_1(\ell) = 12\gamma_E(\gamma_{13}-2)\ell(\ell+1) - 14(\gamma_{13}-2)\ell(\ell+1)S_1(\ell) \tag{C.1}$$

$$+ 6(\gamma_{13}-2)\ell(\ell+1)S_1(\ell+\frac{1}{2}) + 2\left(6\gamma_{13}\ell^2 + (\gamma_{13}-2)(\ell+1)\ell\log(64) - 3\gamma_{13} - 9\ell^2 + 3\ell + 6\right),$$

$$r_2(\ell) = 14(\gamma_{13}-2)\ell S_1(\ell) - 6(\gamma_{13}-2)\ell S_1(\ell+\frac{1}{2}) - 12\gamma_E(\gamma_{13}-2)\ell \tag{C.2}$$

$$- \frac{2\left(6\gamma_{13}\ell^2 - 3\gamma_{13}\ell + (\gamma_{13}-2)(\ell+1)\ell\log(64) - 3\gamma_{13} - 9\ell^2 + 9\ell + 6\right)}{\ell+1}.$$

## C.2   Computing (3.19)

Consider the integral (3.19). The part proportional to $a_1$ can be computed using (A.8), since the spin 0 Mack polynomial is equal to 1. Let us deduce an analytic expression for the nontrivial part

of the integral (3.19), which is the part not proportional to $a_1$. The idea will be to consider (3.19) for noninteger $\ell$ and use the Mellin representation for the Mack polynomial.

$Q_{\ell,m=0}^{\tau=2,d=4}(s)$ has the following Mellin representation

$$Q_{\ell,m=0}^{\tau=2,d=4}(s) = \frac{2^\ell \Gamma(1+\ell)}{(1+\ell)_\ell \Gamma(-\ell)\Gamma(-\frac{s}{2})} \int \frac{ds_1}{2\pi i} \frac{\Gamma(s_1)\Gamma(-\ell-s_1)\Gamma(1+\ell-s_1)\Gamma(-\frac{s}{2}-s_1)(-1)^{-s_1}}{\Gamma(1-s_1)^2}.$$
(C.3)

In the formula above $\ell$ is taken to be non-integer. The contour is bent, in such a way as to pass to the right of the poles of $\Gamma(s_1)$ and to the left of poles of $\Gamma(-\ell-s_1)\Gamma(1+\ell-s_1)\Gamma(-\frac{s}{2}-s_1)$.

We plug this expression in (3.19), exchange the order of integration and evaluate the $s$ integra. This gives

$$\int \frac{ds}{4\pi i}\Gamma(-\frac{s}{2})\Gamma(-\frac{s}{2}-s_1)\Gamma(\frac{s}{2}+1)^2\frac{4+s}{2+s} = \frac{\pi}{\Gamma(s_1)\sin(\pi s_1)}\Big(\frac{\pi^2}{\sin(\pi s_1)^2}-\frac{1}{s_1-1}-\psi^{(1)}(s_1)\Big). \quad (C.4)$$

We obtained this formula by deforming the contour to the right, picking up the poles from $\Gamma(-\frac{s}{2})\Gamma(-\frac{s}{2}-s_1)$ and evaluating the infinite sum over residues. The contour of the integral above is supposed to be bent, separating left from right poles.

Thus, the nontrivial part of (3.19) is equal to

$$\frac{2^\ell \Gamma(1+\ell)}{\pi\Gamma(-\ell)(1+\ell)_\ell}\int \frac{ds_1}{2\pi i}\Gamma(-\ell-s_1)\Gamma(1+\ell-s_1)\Gamma(s_1)^2\sin(\pi s_1)(-1)^{-s_1}\Big(\frac{\pi^2}{\sin(\pi s_1)^2}-\frac{1}{s_1-1}-\psi^{(1)}(s_1)\Big).$$
(C.5)

We evaluate the $s_1$ integral by picking the poles at $s_1 = 0, -1, -2, ....$ We thus transform the $s_1$ integral into an infinite series. Let us take the limit where $\ell$ becomes an integer again. The infinite series must diverge, in order to cancel the $1/\Gamma(-\ell)$. For each integer $\ell$, only $\ell$ terms out of the infinte sum contribute to the divergence. In fact the terms come from just $\Gamma(-\ell-s_1)$. Thus we conclude that the nontrivial part of (3.19) is equal to

$$\frac{2^\ell \Gamma(1+\ell)^2}{(1+\ell)_\ell}\sum_{n=0}^{\ell}\frac{(-1)^n\Gamma(1+\ell+n)}{\Gamma(1+\ell-n)}\Big(\frac{1+n}{\Gamma(2+n)^2}+\frac{\psi^{(1)}(1+n)}{\Gamma(1+n)^2}\Big).$$
(C.6)

We checked that (C.6) is equal to

$$-\frac{\sqrt{\pi}2^{-\ell-1}\Gamma(\ell+1)^2\left((\ell+1)^2\psi^{(1)}\left(\frac{\ell}{2}+1\right)-(\ell+1)^2\psi^{(1)}\left(\frac{\ell+3}{2}\right)-4\right)}{(\ell+1)^2\Gamma\left(\ell+\frac{1}{2}\right)}+\delta_{\ell,0}.$$
(C.7)

that we quoted in the main text.

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
