# Peer review of "Applications of dispersive sum rules: $ε$-expansion and holography"

_SciPost Physics_

## Round 2 · Referee Report · Anonymous (Referee 2) · 2021-3-14

Strengths
1) The paper is well-written and goes through many concrete applications for dispersive sum rules.
2) There are new results for the Wilson-Fisher in the $\epsilon$ expansion, including OPE coefficients at order $\epsilon^4$.
3) The dispersive sum rules are used to derive bounds on AdS EFTs and to determine tree and loop-level anomalous dimensions.
Weaknesses
1) Its less clear which results in the holographic section are new and which ones are a rederivation of previous results.
From my understanding, the results for the one-loop bubble diagram and for the contribution of heavy operators to the sum rule are new and the results for scalar exchange diagrams and the bound on $(\partial \phi)^4$ couplings are rederivations of previous results. It may be useful to clarify this in the text.
Report
The paper is well-written and presents new results on CFTs using dispersive sum rules. I would recommend this paper be published in scipost, with some minor changes.
Requested changes
1) It would useful to define the term "collinear family" and "collinear Mack polynomials". The authors are referring to the contribution of the leading SL(2,R) primaries, although this is not defined in the text.
2) At one-loop and in integer dimensions the bubble diagram has UV divergences. If possible, it would be interesting if the authors could explain how these UV divergences appear using their method.

---

## Round 2 · Referee Report · Anonymous (Referee 1) · 2021-3-27

Strengths
1. Compact and clear.
2. Useful new results.
Weaknesses
1. Unnecessary paragraph breaks after equations, punctuation might help.
2. The appendices are a bit disjoint.
Report
This paper provides interesting applications of Mellin space dispersion relations. It fits nicely into recent work developing this toolkit.
I would recommend this for publication in Sci Post.
Requested changes
It's fine as is. If the authors are going to make the changes recommended by the other reviewer they may want to take into account the comments above about the appendices, equation punctuations, and paragraph breaks, as well as standardize the spelling of Fisher's name.

---

## Editorial Decision

unknown